# ResQ: Mixed-Precision Quantization of Large Language Models with Low-Rank Residuals

Utkarsh Saxena [1]   Sayeh Sharify [2]   Kaushik Roy [1]   Xin Wang [2]

## Abstract

Post-training quantization (PTQ) of large language models (LLMs) holds the promise in reducing the prohibitive computational cost at inference time. Quantization of all weight, activation and key-value (KV) cache tensors to 4-bit without significantly degrading generalizability is challenging, due to the high quantization error caused by extreme outliers in activations. To tackle this problem, we propose *ResQ*, a PTQ method that pushes further the state-of-the-art. By means of principal component analysis (PCA), it identifies a low-rank subspace (in practice $1/8$ of the hidden dimension) in which activation variances are highest, and keep the coefficients within this subspace in high precision, e.g. 8-bit, while quantizing the rest to 4-bit. Within each subspace, invariant random rotation is applied to further suppress outliers. We show that this is a provably optimal mixed precision quantization scheme that minimizes error. With the Llama and Qwen2.5 families of models, we demonstrate that ResQ outperforms recent uniform and mixed precision PTQ methods on a variety of benchmarks, achieving up to 33% lower perplexity on Wikitext than the next best method *SpinQuant*, and upto $5\times$ speedup over 16-bit baseline. Code is available here.[1]

## 1. Introduction

Growing capabilities of large language models (LLMs) come with an increasing computational cost at inference time. LLM inference has two distinct stages: *prefilling*, which processes the input prompt and populates the internal state called KV (key-value) cache, and *generation*, where tokens are generated autoregressively. The prefilling stage is compute-bound, requiring trillions of floating-point operations (FLOPs), whereas the generation stage is memory-bound due to iterative accesses and updates of the KV cache. These high computational costs are further amplified by modern LLMs' large sizes – some exceeding 400 billion parameters – and the increasingly long context lengths that necessitates large KV caches.

Quantization algorithms are powerful and principled approaches to address the immense computational demands of LLMs at both stages of inference. Quantization of weights reduces parameter storage, KV cache quantization lowers memory usage of KV cache during generation, whereas activation quantization decreases the complexity of floating-point operation. However, effective low-precision quantization is difficult due to large outliers in activations, which can be $\sim 20\times$ larger than other values (Dettmers et al., 2022). While post-training methods like KIVI (Liu et al., 2024c) and KVQuant (Hooper et al., 2024) achieve 2-bit KV cache quantization, and techniques like GPTQ (Frantar et al., 2023) and AWQ (Lin et al., 2024c) optimize very low-precision weights, quantizing activations below 8-bit precision remains an open challenge.

Recent LLM activation quantization methods feature two useful strategies: *differential treatment of outliers* retain outlier channels in high precision, leading to mixed-precision quantization (e.g., Dettmers et al. 2022; Zhao et al. 2024; Ashkboos et al. 2024b; Figure 1a), whereas *invariant random rotation* suppress outliers, leading to less difficult uniform low-precision quantization (e.g., Ashkboos et al. 2024c; Liu et al. 2025; Figure 1b). Both reduce quantization error and improve signal-to-quantization-noise ratio (Figure 1d,e) locally; yet a notable model performance gap persists from the 16-bit baseline. For example, SpinQuant (Liu et al., 2025) at 4-bit, applied to `Meta-Llama-3-8B` (Meta, 2024b), exhibits $\sim 20\%$ higher perplexity than the 16-bit floating point baseline, even after nontrivial optimization.

To mend this gap, we introduce *ResQ*, a novel PTQ method that combines the strengths of both aforementioned strategies and thereby improve model efficiency with aggressive 4-bit quantization of activation, weight, and KV cache. Specif-

[1]Department of Electrical and Computer Engineering, Purdue University, West Lafayette, USA [2]d-Matrix, Santa Clara, USA. Correspondence to: Utkarsh Saxena <saxenau@purdue.edu>.

*Proceedings of the 42nd International Conference on Machine Learning*, Vancouver, Canada. PMLR 267, 2025. Copyright 2025 by the author(s).

[1]https://github.com/utkarsh-dmx/project-resq

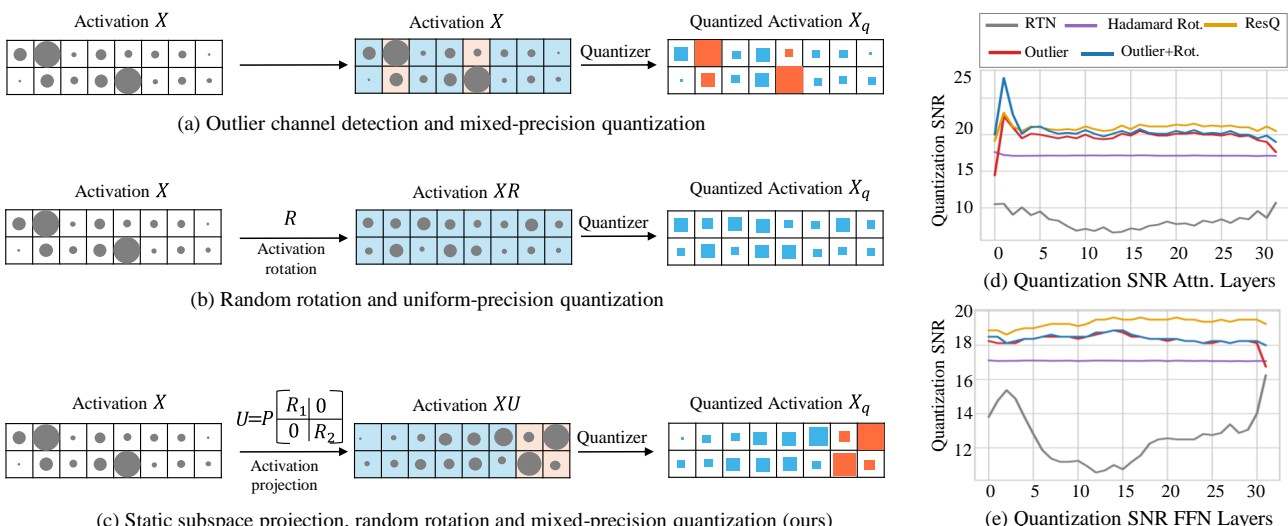

(a) Outlier channel detection and mixed-precision quantization

(b) Random rotation and uniform-precision quantization

(c) Static subspace projection, random rotation and mixed-precision quantization (ours)

(d) Quantization SNR Attn. Layers

(e) Quantization SNR FFN Layers

Figure 1: (a)-(c) Different approaches to quantization including ResQ. Symbol sizes represent magnitudes of values and colors indicate precisions of quantization (blue: low precision, orange: high precision). (d)-(e) Quantization SNR comparison of ResQ with other baselines.

ically, by means of principal component analysis (PCA), we first identify a low-rank subspace that captures highest variances in activation, and mark the coefficients along this subspace for high-precision (8-bit) and the complement subspace for low-precision (4-bit) quantization. Then, ResQ employs invariant random rotations within each subspace before quantization to further suppress outliers (Figure 1c,d,e). We prove that the above treatment minimizes quantization error. Similar to SpinQuant, most projection matrices can be fused into adjacent weights, leading to minimal runtime computational overhead (Section 4.3). Furthermore, ResQ can be applied to KV cache quantization as well, and can be combined with GPTQ (Frantar et al., 2023), resulting in even better generalizing LLMs.

Outlier-based and rotation-based quantization methods can be combined. For example, high-precision outliers could be detected by $\ell_\infty$-norm similar to QUIK (Ashkboos et al., 2024b), and random rotations applied within both high- and low-precision quantization groups, as in QuaRot (Ashkboos et al., 2024c). These methods fare less well than ResQ (Figure 1d,e) in practice, in support of ResQ's provably optimal treatment of outlier quantization. When quantizing weight, activation and KV cache to 4-bit with only $1/8$ channels in 8-bit, ResQ achieves 4-33% lower perplexity on Wikitext and 0.1-5.4% 0-shot accuracy improvements over SpinQuant (Liu et al., 2025), the best in practice so far. Unlike SpinQuant, ResQ does not require gradient-based optimization, making it a less demanding and faster PTQ solution. Furthermore, tuning the rank $r$ of ResQ gives rise to Pareto-optimal solutions as a tradeoff between efficiency and accuracy. We claim the following contributions.

1. We propose ResQ, a mixed precision weight, activation, and KV cache quantization method by keeping low-rank, high-variance components in high precision, in combination with random rotation-induced outlier suppression.

2. We theoretically analyze the projection matrices in ResQ and show that using PCA-based projections minimizes quantization error.

3. We conduct extensive experiments on various models and language tasks and show that ResQ outperforms related state-of-the-art approaches.

4. We develop CUDA kernels and achieve runtime speedup on NVIDIA GPUs with our quantized models.

## 2. Prior Work

### 2.1. Quantization of LLMs

Quantization reduces model size and accelerates inference by lowering neural network bit precision (Choi et al., 2018; Hubara et al., 2021; Yao et al., 2022; Gholami et al., 2022; Xi et al., 2023; Park et al., 2024). It is broadly categorized into two categories: *uniform precision quantization* (UPQ) and *mixed precision quantization* (MPQ). **Uniform precision quantization (UPQ)** applies the same bit-width across all layers, simplifying implementation but neglecting layer-specific sensitivity to quantization. **Weight-only UPQ** methods reduce storage by compressing weights, using techniques like Hessian-guided rounding (GPTQ, Frantar et al. 2023), adaptive rounding (QuIP, Chee et al. 2023), channel-wise scaling (AWQ, Lin et al. 2024c), and multi-codebook

quantization (AQLM, Egiazarian et al. 2024). However, these methods struggle with batch processing due to significant activation memory overhead. **Weight-activation UPQ** compresses both weights and activations to address this. Methods such as SmoothQuant (Xiao et al., 2023) and OmniQuant (Shao et al., 2024) scale activations and weights to handle outliers, while RPTQ (Yuan et al., 2023a), QLLM (Liu et al., 2024a), and QServe (Lin et al., 2025) employ channel-level strategies like clustering and reordering. Rotation-based methods such as QuaRot (Ashkboos et al., 2024c), SpinQuant (Liu et al., 2025) and DuQuant (Lin et al., 2024b) further enhance robustness in low-precision scenarios. **KV cache UPQ** reduces memory for large batches or long contexts. FlexGen (Sheng et al., 2023) employs 4-bit quantization and memory offloading, while KIVI (Liu et al., 2024c) uses asymmetric 2-bit quantization for compression, enabling efficient inference.

**Mixed precision quantization (MPQ)** optimizes bit-widths by adapting to the sensitivity of weights and activations, achieving better accuracy than UPQ at similar compression rates. *Our proposed method, ResQ, follows the MPQ approach.* **Weight-only MPQ** has advanced efficiency for memory-bound applications with minimal activation demands. Methods like OWQ (Lee et al., 2024) and SpQR (Dettmers et al., 2024) mitigate activation outliers' impact by retaining critical features in full precision, while SqueezeLLM (Kim et al., 2024) employs Dense-and-Sparse decomposition to efficiently store sensitive weights. **Weight-activation MPQ** enhances efficiency by addressing activation outliers (e.g. (Guan et al., 2024; Zeng et al., 2025)). Methods like LLM.int8() (Dettmers et al., 2022) and QUIK (Ashkboos et al., 2024b) preserve critical activations with mixed or low-precision decompositions, while Atom (Zhao et al., 2024) and SliM-LLM (Huang et al., 2024) optimize quantization through channel reordering and salience-driven bit allocation. **KV cache MPQ** reduces memory usage while preserving precision for critical tokens using techniques like non-uniform quantization, importance-aware precision, and salient token compression (Hooper et al., 2024; Yang et al., 2024b; Dong et al., 2024; He et al., 2024). Alternatively, GEAR quantizes all tokens' KV cache and maintains low-rank quantization error (Kang et al., 2024).

### 2.2. Low-rank Decomposition

Low-rank decomposition techniques have been widely used in model compression, reducing dimensionality while maintaining performance. For instance, SliceGPT (Ashkboos et al., 2024a) projects weight matrices onto principal components for sparsification, while ESPACE (Sakr & Khailany, 2024) reduces activation dimensionality via pre-calibrated projections, achieving inference-time efficiency. Similarly, ASVD (Yuan et al., 2023b) introduces an activation-aware

decomposition method that incorporates activation distributions into weight decomposition. Additionally, low-rank decomposition can be applied to reduce KV cache size. For example, Eigen Attention (Saxena et al., 2024) and ASVD (Yuan et al., 2023b) employ low-rank approximations to reduce memory usage in KV caches during attention operations. PALU (Chang et al., 2024) introduces learnable projections to adaptively compress KV caches based on the compression budget. Finally, Matryoshka KV Cache (Lin et al., 2024a) refines this with hierarchical orthogonal projections and knowledge distillation.

## 3. Quantization

Quantization of weight, activation or KV cache involves converting component elements to low precision so that they can be represented using fewer bits for more efficient compute and storage. The $N$-bit integer quantization and dequantization process on matrix $\boldsymbol{X}$ is given as

$$Q_N(\boldsymbol{X}) = \left\lfloor \frac{\boldsymbol{X} - z_X}{s_X} \right\rceil \cdot s_X + z_X, \qquad (1)$$

where $\lfloor \cdot \rceil$ is a round-and-clip function; $s_X$ and $z_X$ the scale and zero-point; $z_X = 0$, $s_X = \frac{\max(|\boldsymbol{X}|)}{2^{N-1}-1}$ for symmetric quantization or $z_X = \min(\boldsymbol{X})$, $s_X = \frac{\max(\boldsymbol{X})-\min(\boldsymbol{X})}{2^N-1}$ for asymmetric quantization.

## 4. ResQ

In this section, we introduce ResQ, a mixed-precision quantization approach that projects weights, activations, and the KV cache into an orthogonal space, retaining the low-rank components in high precision (8-bit) and the rest in low precision. We describe the quantization scheme, the generation of the basis space, provide theoretical guarantees, and outline end-to-end LLM inference deployment procedure.

### 4.1. Quantization Scheme

Given input activation $\boldsymbol{X} \in \mathbb{R}^{n \times d}$ and weight $\boldsymbol{W} \in \mathbb{R}^{d \times d}$, they are first projected onto an orthogonal basis defined by the vectors $\boldsymbol{U} \in \mathbb{R}^{d \times d}$. The coefficients of the projections along this basis are then subject to quantization. We seek to quantize some coefficients along certain bases at high precision while those remaining at low precision. Within $\mathbb{R}^d$, denote bases of a low-rank space of high-precision components by $\boldsymbol{U}_h \in \mathbb{R}^{d \times r}$ and those of its complementary subspace of low-precision components by $\boldsymbol{U}_l \in \mathbb{R}^{d \times (d-r)}$. The rank $r$ controls the amount of components in high precision (in practice we typically choose $r = d/8$). We have $\boldsymbol{U}_h \boldsymbol{U}_h^\top + \boldsymbol{U}_l \boldsymbol{U}_l^\top = \boldsymbol{U} \boldsymbol{U}^\top = \boldsymbol{I}$ because $\boldsymbol{U}$ is orthogonal. The quantized activation $\boldsymbol{X}_q$ is thusly

$$\boldsymbol{X}_q = Q(\boldsymbol{X}\boldsymbol{U}) = [Q_L(\boldsymbol{X}\boldsymbol{U}_l) \ \ Q_H(\boldsymbol{X}\boldsymbol{U}_h)]. \qquad (2)$$

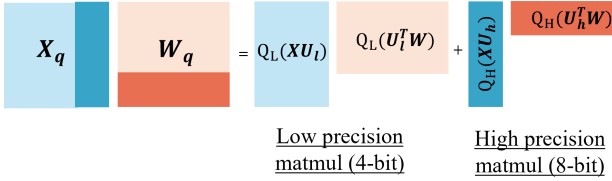

Figure 2: Matrix multiplication with mixed precision operands

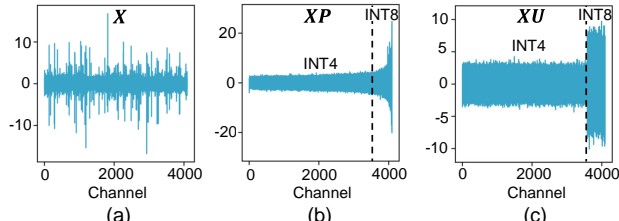

Figure 3: Activation distribution of the baseline and applying the projection matrices.

Similarly, quantized weights $\boldsymbol{W}_q$ is obtained by projecting the inputs space of weights by $\boldsymbol{U}^\top$ and quantizing the coefficients,

$$\boldsymbol{W}_q = Q(\boldsymbol{U}^\top \boldsymbol{W}) = \begin{bmatrix} Q_L(\boldsymbol{U}_l^\top \boldsymbol{W}) \\ Q_H(\boldsymbol{U}_h^\top \boldsymbol{W}) \end{bmatrix}. \tag{3}$$

And the output of the layer is,

$$\begin{aligned} \boldsymbol{X}_q \boldsymbol{W}_q &= Q_L(\boldsymbol{X}\boldsymbol{U}_l) Q_L(\boldsymbol{U}_l^\top \boldsymbol{W}) \\ &+ Q_H(\boldsymbol{X}\boldsymbol{U}_h) Q_H(\boldsymbol{U}_h^\top \boldsymbol{W}). \end{aligned} \tag{4}$$

We make two observations due to orthogonality. First, *the introduction of the projections do not alter the output of the model at infinite precision*. This means that, if quantization operation is removed from Equation 4, the layer output is numerically invariant. Second, *multiplication between low- and high-precision components vanishes (Figure 2)*. This is efficient because only hardware kernels for quantized GEMM between operands of same precision are required.

### 4.2. Projections and Optimality Thereof

Intuitively, the orthogonal basis vectors $\boldsymbol{U}$ should have two properties: (1) the low-rank space for high-precision quantization should capture the more important components, and (2) quantization error in both high- and low-precision groups should be minimized. We construct $\boldsymbol{U}$ as a combination of two rotation matrices serving both objectives respectively. We write $\boldsymbol{U}_i = \boldsymbol{P}_i \boldsymbol{R}_i, i \in \{h, l\}$. Therefore,

$$\boldsymbol{U} = \boldsymbol{P}\boldsymbol{R} = [\boldsymbol{P}_l\,\boldsymbol{P}_h] \begin{bmatrix} \boldsymbol{R}_l & \boldsymbol{0} \\ \boldsymbol{0} & \boldsymbol{R}_h \end{bmatrix}, \tag{5}$$

where, $\boldsymbol{P}_l, \boldsymbol{R}_l \in \mathbb{R}^{d \times (d-r)}, \boldsymbol{P}_h, \boldsymbol{R}_h \in \mathbb{R}^{d \times r}$. Inspired by prior work (Ashkboos et al., 2024c; Chee et al., 2023), we make $\boldsymbol{R}_l, \boldsymbol{R}_h$ random orthogonal matrices because random rotation reduces outliers, making the rotated matrices easier to quantize. Furthermore, projection with a random orthogonal matrix increases Gaussianity of activations and weights within high- and low-precision groups, due to Lemma 4.1, conducive to the quantizations applied to these groups.

**Lemma 4.1.** *By Central Limit Theorem, the distribution after multiplication with random orthogonal matrix is approximately Gaussian (Tseng et al., 2024).*

To determine $\boldsymbol{P}$, we minimize the activation quantization error $\|\boldsymbol{X} - \boldsymbol{X}_q\|_F$. For activations quantized according to Equation 2, we have,

$$\begin{aligned} \|\boldsymbol{X} - \boldsymbol{X}_q\|_F &= \|\boldsymbol{X}\boldsymbol{U}_l - Q_L(\boldsymbol{X}\boldsymbol{U}_l)\|_F \\ &+ \|\boldsymbol{X}\boldsymbol{U}_h - Q_H(\boldsymbol{X}\boldsymbol{U}_h)\|_F. \end{aligned} \tag{6}$$

**Theorem 4.2.** *For any matrix $\boldsymbol{X}$ quantized to $\boldsymbol{X}_q$ according to method described in Equation 2, assuming the values to be quantized in $\boldsymbol{X}$ are normally distributed, we have*

$$\begin{aligned} \mathbb{E}\|\boldsymbol{X} - \boldsymbol{X}_q\|_F &\leq \frac{\sqrt{\pi \log(d-r)}}{2^{L-1} - 1} \mathbb{E}\|\boldsymbol{X}\|_F \\ &- \left[ \frac{\sqrt{\pi \log(d-r)}}{2^{L-1} - 1} - \frac{\sqrt{\pi \log r}}{2^{H-1} - 1} \right] \\ &\mathbb{E}\|\boldsymbol{X}\boldsymbol{P}_h\|_F. \end{aligned} \tag{7}$$

Full proof of Theorem 4.2 is in Appendix A. Theorem 4.2 bounds the quantization error in Equation 6 from above. To lower this upper bound of quantization error is thusly to maximize $\|\boldsymbol{X}\boldsymbol{P}_h\|_F$ which happens when $\boldsymbol{P}_h$ comprises of eigenvectors of the covariance matrix $\boldsymbol{X}\boldsymbol{X}^\top$ with its *largest* eigenvalues. Therefore, the low-rank subspace for high-precision quantization can be obtained by means of PCA, while the subspace for low-precision quantization can be obtained using $\boldsymbol{U}_h\boldsymbol{U}_h^\top + \boldsymbol{U}_l\boldsymbol{U}_l^\top = \boldsymbol{P}_h\boldsymbol{P}_h^\top + \boldsymbol{P}_l\boldsymbol{P}_l^\top = \boldsymbol{I}$ (because $\boldsymbol{R}_i$ is orthogonal). If we construct $\boldsymbol{P}$ by taking eigenvectors of $\boldsymbol{X}\boldsymbol{X}^\top$ arranged in *increasing* order of eigenvalues, the last $r$ columns of such a $\boldsymbol{P}$ would correspond to $\boldsymbol{P}_h$ and the first $d - r$ columns would correspond to $\boldsymbol{P}_l$. The distribution of activation after applying different projection matrices is shown in Figure 3. Projection of activation along $\boldsymbol{P}$ sorts the activation coefficients in increasing order of variance due to increasing eigenvalues of bases vectors. Consequently, the later $r$ channels of the projected activations with higher variance are kept in higher precision. Projection along $\boldsymbol{U} = \boldsymbol{P}\boldsymbol{R}$ smoothes the activations along low precision and high precision groups further reducing quantization error (Figure 3) and improving quantization SNR (Figure 1(d,e)).

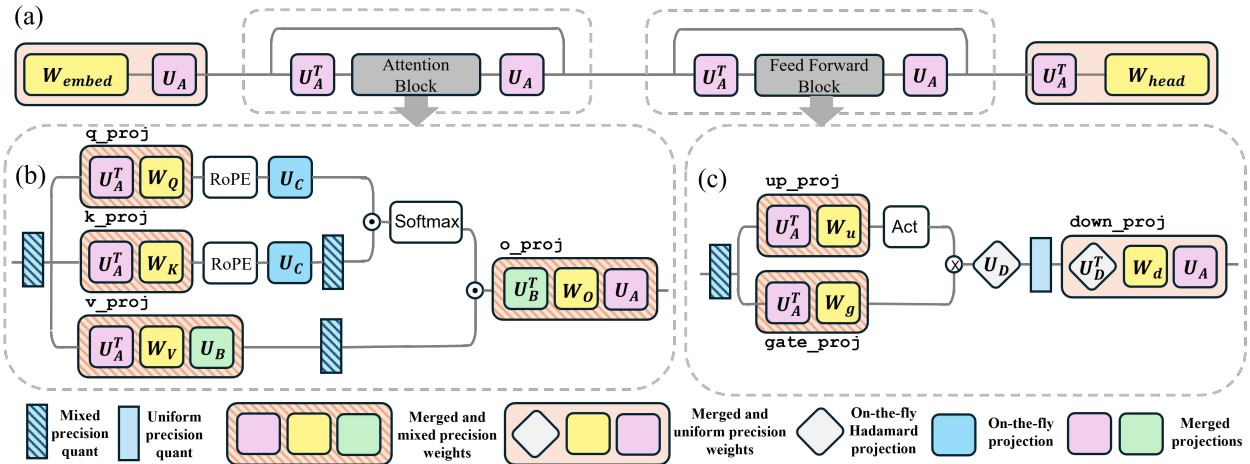

Figure 4: Model inference with ResQ incorporating the projection matrices. (a) $U_A$ modifies the inputs across blocks enabling better quantization. (b) $U_B, U_C$ enables mixed precision quantization of KV cache. (c) $U_D$ projects the activations and weights of down_proj layer.

### 4.3. Inference Computation with Optimized Projections

Once the projection matrices are obtained, the operation in Equation 4 requires multiplying the weights and activations with $U$. Weights can be projected and quantized offline. The projection operation on an activation can be merged to the weight of a previous linear layer. Based on the architecture of decoder based LLMs, we introduce four different kinds of projections (Figure 4) : $U_A \in \mathbb{R}^{d_h \times d_h}$, $U_B, U_C \in \mathbb{R}^{d_{\text{head}} \times d_{\text{head}}}$, $U_D \in \mathbb{R}^{d_{\text{FFN}} \times d_{\text{FFN}}}$ where $d_h$ is hidden dimension of LLM, $d_{\text{head}}$ is the attention head dimension and $d_{\text{FFN}}$ the hidden dimension of feedforward network (FFN).

**Projections at block boundaries** Input activations to attention and FFN are projected via $U_A$. Projection is handled by right-multiplying the weight matrix of final linear layer in each block (o_proj in attention and down_proj in FFN) by $U_A$. Thus, projections of activations is handled at no additional inference cost. To maintain numerical invariance, the first linear layer of each block (q_proj|k_proj|v_proj in attention and up_proj|gate_proj in FFN) is pre multiplied with $U_A^\top$ (Figure 4a). Similarly, the weights of the embedding layer and the final head are modified to manage projection of the residual stream.

**Projections within the attention block** $U_B, U_C$ ensures that activations within attention block are projected (Figure 4b). Post-multiplication of value projection layer by $U_B$ ensures that value vectors in KV cache are projected and quantized optimally. Consequently, the weights of o_proj layer need to be pre multiplied by $U_B^\top$ to ensure numerical invariance. $U_C$ ensures that the quantization of key in KV cache is handled optimally. To achieve that, it is required

to project both the query and key using the same projection matrix $U_C$. The attention dot product remains invariant under projected inputs,

$$q_{\text{proj}} K_{\text{proj}}^\top = (q U_C)(U_C^\top K^\top) = q K^\top, \qquad (8)$$

where $q$ and $K$ are query and key after rotary embedding (RoPE), respectively. Because $U_C$ cannot be merged into the previous linear layer due to presence of RoPE, the projection is explicitly computed at runtime, but made more efficient by applying uniform precision quantization to $U_C$ and corresponding input activations.

**Projections within the feedforward block** $U_D$ ensure improved quantization of activation within FFNs (Figure 4c). $U_D^\top$ is left-multiplied with weights of down_proj, but due to the presence of activation functions within the block, $U_D$ cannot be merged to weights of preceding linear layers and is computed at runtime. $U_D$ is applied to the hidden dimension of the FFNs ($d_{\text{FFN}}$) which is typically $3\times$ to $4\times$ the embedding dimension in most LLMs. In this scenario, matrix multiplication with $U_D$ is extremely expensive in computation and storage. To minimize the overhead, we choose $U_D$ to be a hadamard matrix to leverage fast and efficient hadamard transform kernel. And, we choose weights and activations for down_proj layer to be uniformly quantized to low precision.

## 5. Experiments

### 5.1. Setup

**Models, tasks, datasets and baselines** We conduct experiments on Llama 2 (Touvron et al., 2023), Llama 3 (Meta, 2024b), and the recently released Llama 3.2 (Meta, 2024a)

Table 1: Comparison of perplexity score on Wikitext, average 0-shot common sense reasoning accuracy and average 0-shot MMLU accuracy. Results of all techniques were obtained using their official codebase. Our work ResQ and QUIK (Ashkboos et al., 2024b) keep $1/8$ channels in 8-bit and remaining in 4-bit for W/A/KV = 4.5-bit. Other baselines are uniformly quantized to W/A/KV = 4-bit. All techniques except RTN employ GPTQ (Frantar et al., 2023) for weight quantization. $\uparrow$ higher is better, $\downarrow$: lower is better. Full results in Appendix D, Tables 10 and 11.

| Family | Method | W/A/KV | Meta-Llama-3-8B | | | Meta-Llama-3-70B | | |
| --- | --- | --- | --- | --- | --- | --- | --- | --- |
| | | | Wiki ($\downarrow$) | Avg. 0-shot ($\uparrow$) | MMLU ($\uparrow$) | Wiki ($\downarrow$) | Avg. 0-shot ($\uparrow$) | MMLU ($\uparrow$) |
| Llama 3 | 16-bit baseline | 16/16/16 | 6.1 | 67.1 | 63.1 | 2.9 | 73.1 | 75.9 |
| | RTN | 4/4/4 | 218.9 | 39.3 | 23.6 | 452.7 | 45.5 | 23.2 |
| | GPTQ | 4/4/4 | 166.3 | 39.8 | 23.3 | 11.6e3 | 34.9 | 25.5 |
| | SmoothQuant+ | 4/4/4 | 78.2 | 42.5 | 24.7 | - | - | - |
| | QUIK | 4.5/4.5/4.5 | 14.2 | 51.6 | 32.7 | 8.0 | 58.2 | 51.1 |
| | QuaRot | 4/4/4 | 7.8 | 62.1 | 53.2 | 5.7 | 67.6 | 65.3 |
| | SpinQuant | 4/4/4 | 7.4 | 63.8 | 56.2 | 6.2 | 65.7 | 59.4 |
| | ResQ | 4.5/4.5/4.5 | **7.1** | **63.9** | **57.2** | **4.1** | **71.1** | **73.9** |
| Family | Method | W/A/KV | Llama-3.2-1B | | | Llama-3.2-3B | | |
| | | | Wiki ($\downarrow$) | Avg. 0-shot ($\uparrow$) | MMLU ($\uparrow$) | Wiki ($\downarrow$) | Avg. 0-shot ($\uparrow$) | MMLU ($\uparrow$) |
| Llama 3.2 | 16-bit baseline | 16/16/16 | 9.8 | 54.9 | 36.9 | 7.8 | 62.7 | 54.8 |
| | RTN | 4/4/4 | 329.1 | 38.1 | 23.8 | 268.8 | 38.7 | 25.7 |
| | GPTQ | 4/4/4 | 108.9 | 38.0 | 24.9 | 178.3 | 40.3 | 24.8 |
| | SmoothQuant+ | 4/4/4 | 228.9 | 38.0 | 24.1 | 96.1 | 39.0 | 25.9 |
| | QUIK | 4.5/4.5/4.5 | 21.8 | 44.3 | 25.1 | 15.8 | 48.8 | 31.1 |
| | QuaRot | 4/4/4 | 14.3 | 49.0 | 25.5 | 10.1 | 56.1 | 42.0 |
| | SpinQuant | 4/4/4 | 13.6 | 48.8 | 25.6 | 9.2 | 57.9 | 44.2 |
| | ResQ | 4.5/4.5/4.5 | **12.4** | **50.1** | **29.4** | **8.8** | **59.0** | **49.8** |
| Family | Method | W/A/KV | Qwen2.5-3B | | | Qwen2.5-72B | | |
| | | | Wiki ($\downarrow$) | Avg. 0-shot ($\uparrow$) | MMLU ($\uparrow$) | Wiki ($\downarrow$) | Avg. 0-shot ($\uparrow$) | MMLU ($\uparrow$) |
| Qwen2.5 | 16-bit baseline | 16/16/16 | 8.0 | 63.8 | 66.1 | 3.9 | 73.4 | 84.3 |
| | RTN | 4/4/4 | 39033.0 | 35.1 | 23.4 | 45412.7 | 34.3 | 24.0 |
| | GPTQ | 4/4/4 | 9977.8 | 35.1 | 23.2 | 37967.2 | 34.5 | 23.3 |
| | SmoothQuant+ | 4/4/4 | 73306.7 | 34.8 | 23.9 | - | - | - |
| | QUIK | 4.5/4.5/4.5 | 15.5 | 51.2 | 39.4 | 8.3 | 61.9 | 69.3 |
| | QuaRot | 4/4/4 | 68.8 | 47.7 | 28.9 | 4.9 | 70.3 | 80.1 |
| | ResQ | 4.5/4.5/4.5 | **9.0** | **61.1** | **61.2** | **4.6** | **72.0** | **81.5** |

Table 2: Comparison of performance of quantization approaches on generative tasks. Our work ResQ and QUIK (Ashkboos et al., 2024b) keep $1/8$ of channels in 8-bit and remaining in 4-bit for W/A/KV = 4.5-bit. Other baselines are uniformly quantized to W/A/KV = 4-bit.

| Model | Method | W/A/KV | GSM8K 5-shot ($\uparrow$) | | LongBench ($\uparrow$) | | |
| --- | --- | --- | --- | --- | --- | --- | --- |
| | | | flexible extract | strict match | qmsum | samsum | repobench-p |
| Meta-Llama-3-8B | 16-bit baseline | 16/16/16 | 51.0 | 50.6 | 23.9 | 44.8 | 66.4 |
| | QUIK | 4.5/4.5/4.5 | 2.3 | 0.0 | 10.5 | 25.2 | 37.6 |
| | QuaRot | 4/4/4 | 27.6 | 27.1 | 22.0 | 43.8 | 60.6 |
| | SpinQuant | 4/4/4 | 29.8 | 29.6 | 23.0 | 43.9 | **62.6** |
| | ResQ | 4.5/4.5/4.5 | **33.6** | **33.2** | **23.1** | **44.1** | 62.3 |
| Llama-3.2-3B | 16-bit baseline | 16/16/16 | 25.1 | 24.9 | 23.1 | 43.0 | 64.4 |
| | QUIK | 4.5/4.5/4.5 | 2.5 | 0.0 | 15.9 | 31.7 | 30.9 |
| | QuaRot | 4/4/4 | 10.1 | 9.1 | 20.6 | 39.5 | 56.8 |
| | SpinQuant | 4/4/4 | 11.6 | 11.4 | **21.7** | 41.9 | 59.1 |
| | ResQ | 4.5/4.5/4.5 | **17.1** | **16.7** | **21.7** | **43.0** | **61.5** |

and Qwen2.5 (Yang et al., 2024a) models. We also include multi-modal language models belonging to Qwen2 VL family (Wang et al., 2024) for our evaluations. We benchmark our approach against GPTQ (Frantar et al., 2023), QuaRot (Ashkboos et al., 2024c), QUIK (Ashkboos et al., 2024b), SpinQuant (Liu et al., 2025) and SmoothQuant+, a stronger baseline created by combining SmoothQuant (Xiao et al., 2023) with GPTQ following Sharify et al. 2024. We evalu-

ate the quantization approaches on a range of tasks which measure the *language modeling ability*: perplexity on Wikitext (Merity et al., 2017), *common sense reasoning ability*: average 0-shot accuracy on Arc-c/e (Clark et al., 2018), BoolQ (Clark et al., 2019), HellaSwag (Zellers et al., 2019), Openbook QA (Mihaylov et al., 2018), PIQA (Bisk et al., 2020), SIQA (Sap et al., 2019), WinoGrande (Sakaguchi et al., 2021), *language understanding*: 0-shot accuracy on

Table 3: MMMU accuracy (higher is better) of vision language models when quantized using various approaches. For 4-bit data structures, our work ResQ and QUIK (Ashkboos et al., 2024b) keep ¹/₈ of channels in 8-bit.

| W/A/KV (bit) | Method | Model | |
|---|---|---|---|
| | | Qwen2-VL -2B-Instruct | Qwen2-VL -7B-Instruct |
| 16/16/16 | Baseline | 39.6 | 51.6 |
| 4/4/4 | RTN | 25.0 | 26.7 |
| | GPTQ | 27.7 | 24.9 |
| | QuaRot | 24.0 | 24.5 |
| 4.5/4.5/4.5 | QUIK | 26.3 | 28.9 |
| | ResQ | **29.7** | **47.0** |
| 4/8/4 | RTN | 24.9 | 25.2 |
| | GPTQ | 23.4 | 24.3 |
| | QuaRot | 26.5 | 24.5 |
| 4.5/8/4.5 | QUIK | 28.4 | 26.4 |
| | ResQ | **34.0** | **48.8** |

Table 4: Wikitext perplexity comparison of ResQ and baseline which keeps ¹/₈ channels with high $l_\infty$-norm in 8-bit (and remaining in low precision) and uses rotation to reduce quantization error within high precision and low precision groups.

| W/A/KV (bit) | Method | Meta-Llama -3-8B | Llama -3.2-3B |
|---|---|---|---|
| 4.5/4.5/4.5 | outlier+rot | 7.2 | 9.0 |
| | ResQ | **7.1** | **8.8** |
| 4.5/8/4.5 | outlier+rot | 6.6 | 8.3 |
| | ResQ | **6.5** | **8.2** |
| 3.6/8/3.6 | outlier+rot | 7.7 | 9.9 |
| | ResQ | **7.5** | **9.8** |
| 2.75/8/8 | outlier+rot | 12.6 | 16.0 |
| | ResQ | **12.1** | **15.7** |
| 3.6/3.6/3.6 | outlier+rot | 14.7 | 18.7 |
| | ResQ | **14.6** | **17.5** |

MMLU (Hendrycks et al., 2021), *mathematical understanding*: 5-shot GSM8K (Cobbe et al., 2021), *dialogue summarization*: samsum (Gliwa et al., 2019) and qmsum (Zhong et al., 2021) from LongBench (Bai et al., 2024), *code completion*: repobench-p (Liu et al., 2024b) from LongBench, and *multi-modal understanding*: MMMU (Yue et al., 2024).

**Implementation details** We implement ResQ using the HuggingFace Transformers library (Wolf et al., 2020) with PyTorch (Paszke et al., 2019). We share a single $U_A$ across all layers, while $U_B$, $U_C$ and $U_D$ are generated per layer. Following SpinQuant (Liu et al., 2025), we use per-token asymmetric quantization for activations, per-channel symmetric quantization for weights, and per-head asymmetric quantization for the KV cache. We fuse the projection matrices $U_A, U_B, U_D$ into weights and apply GPTQ (Frantar et al., 2023) for weight quantization. To efficiently implement on-the-fly projections, $U_D$ is a Hadamard matrix and $U_C$ and its activations are quantized to 8-bit. The entire process, including obtaining projections and quantization, runs on a single NVIDIA A100 GPU; for Meta-Llama-3-8B, it takes 35 minutes. Additional details are in Appendix C.

## 5.2. Main Results

**Language modeling, understanding, and reasoning tasks** We evaluate ResQ on tasks that test language modelling ability (perplexity on Wikitext), common sense reasoning ability (average 0-shot accuracy on the eight tasks listed in section 5.1) and language understanding (average 0-shot accuracy on MMLU). The results are presented in Table 1. We see that ResQ reduces the gap to 16-bit performance and outperforms the quantization baselines across all tasks on all models. Particularly, on Llama 3/3.2 family of models, ResQ outperforms SpinQuant by achieving 4-33% lower Wikitext perplexity, 0.1-5.4% better average 0-shot accuracy and a 1-14.5% better accuracy on MMLU benchmark without any additional training. For the Qwen-2.5 model family, all

other baselines fail to achieve competitive results, and ResQ significantly outperforms them. Compared with QUIK, another mixed precision quantization approach, ResQ achieves 42-50% better Wikitext perplexity, 5.8-12.3% better average zero shot accuracy and 4.3-24.5% better MMLU accuracy over all models. Complete results on Llama and Qwen2.5 family of models are provided in Appendix D. Additionally, we provide comparison between baselines at **W/A/KV = 4/8/4** bits for Llama families in Appendix E. Among all the set of results, ResQ maintains superior performance.

**Generative tasks** We also test ResQ on tasks that require auto-regressive token generation including the GSM8K mathematical understanding benchmark, dialogue summarization benchmarks (qmsum and samsum) and code completion benchmark (repobench-p, Table 2). The goal of choosing these tasks is to evaluate the generation ability on a wide variety of domains. On the challenging GSM8K benchmark where QUIK fails to produce meaningful results, ResQ outperforms SpinQuant by 3.8% and 5.5% on the 8B and 3B parameter model respectively, closing the gap to the 16-bit baseline. On LongBench evaluation tasks, ResQ demonstrates competitive performance and outperforms SpinQuant without any additional training.

**Multi-modal understanding** We benchmark the quantization approaches on vision language models (VLMs) by quantizing Qwen2 VL family and evaluating their performance on MMMU (Table 3, Yue et al. 2024). Only the language model is quantized while the vision encoder remains in 16-bit as the language model has many more parameters (over 10× for Qwen2-VL-7B-Instruct). ResQ outperforms baselines on both 2B and 7B models, achieving superior accuracy and demonstrating its generalizability. Results for individual MMMU tasks are provided in Appendix F.

**Comparison against outliers with rotation baseline** A stronger baseline can be created combining existing quanti-

Table 5: Comparison of perplexity score on Wikitext, average 0-shot common sense reasoning accuracy and average MMLU accuracy at W/A/KV = 4-bit. ResQ keeps $1/8$ channels with high eigen value in 6-bit, $1/8$ channels with low eigen value in 2-bit and the rest in 4-bit for average bit-width of 4-bit. Complete results are provided in Appendix G.

| Model | Method | Wiki ($\downarrow$) | Avg. 0-shot ($\uparrow$) | MMLU($\uparrow$) |
|---|---|---|---|---|
| Qwen2.5-3B | QuaRot | 68.8 | 47.7 | 28.9 |
| | SpinQuant | 70.6 | 48.6 | 32.8 |
| | ResQ | **9.8** | **59.1** | **52.2** |
| Qwen2.5-7B | QuaRot | 4e3 | 38.4 | 24.1 |
| | SpinQuant | 3e3 | 38.6 | 24.3 |
| | ResQ | **34.2** | **56.2** | **58.0** |
| Qwen2.5-14B | QuaRot | 6.8 | 67.1 | 70.9 |
| | SpinQuant | 6.6 | 67.4 | 70.1 |
| | ResQ | **6.5** | **67.5** | **71.3** |
| Qwen2.5-32B | QuaRot | 6.1 | 67.8 | 77.0 |
| | SpinQuant | 6.0 | 67.9 | 77.6 |
| | ResQ | **5.9** | **69.1** | **77.9** |
| Qwen2.5-72B | QuaRot | **4.9** | 70.3 | **80.1** |
| | ResQ | **4.9** | **71.1** | **80.1** |

Table 6: Impact of different projections in ResQ. Evaluated by removing components and observing Wikitext perplexity.

| | ResQ | Removed Projections | | | | |
|---|---|---|---|---|---|---|
| | | $U_D$ | $U_A$ | $U_B$ | $U_C$ | $U_C$, $U_B$ |
| Llama-2-7b-hf | **5.8** | 1550 | 2500 | 5.8 | 5.9 | 5.9 |
| Meta-Llama-3-8B | **7.1** | 1607 | 37.4 | 7.2 | 7.3 | 7.4 |
| Llama-3.2-3B | **8.8** | 279.2 | 39.0 | 9.0 | 9.2 | 9.4 |

zation approaches. Like QUIK, one can find channels which consistently contain outliers and keep them in 8-bit while keep the remaining channels in low precision. And, the quantization of high/low precision groups can be improved using random rotations introduced in QuaRot. Compared with such a baseline which keeps channels with high $l_\infty$-norm in 8-bit, ResQ's unique approach involves keeping coefficients along bases with high eigenvalues in 8-bit. We see in Table 4 that ResQ consistently outperforms such a strong baseline across various precisions of W/A/KV highlighting ResQ's PCA driven theoretically optimal approach of choosing high precision components.

**Iso-bitwidth comparison** We also perform iso-bitwidth comparison of ResQ with SpinQuant and QuaRot at W/A/KV of 4-bit. To enable 4-bits with ResQ, we keep $1/8$ channels corresponding to highest eigen values in $P$ in 6-bit, $1/8$ channels corresponding to lowest eigen values in $P$ in 2-bit and remaining in 4-bit. Within each quantization group, we apply random orthogonal rotations to minimize quantization error. As shown in Table 5, even at same bitwidth of 4-bit, ResQ achieves improved performance on Qwen2.5 family of models. Complete results are provided in Table 14.

**Training rotation matrix $R$** To further improve the per-

formance of ResQ, the random rotation matrix $R$ can be optimized to minimize final task loss similar to SpinQuant (Liu et al., 2025) albeit at higher computational cost for quantizing the model. We keep identical training hyperparameters as SpinQuant, and learn the rotation $R$ for models upto 8B parameters. The evaluation results are provided in Table 15 in Appendix H. We see upto 5% improvement in Wikitext perplexity, upto 0.6% increase in average 0-shot reasoning accuracy and upto 1.1% increase in MMLU accuracy.

### 5.3. Hardware Performance

We implement the mixed-precision quantization using CUDA 11.8 and PyTorch. The weights are quantized into `INT4` and `INT8` components offline while online quantization of activations into `INT4` and `INT8` components is handled in a single kernel call. We use `CUTLASS` (Thakkar et al., 2023) to perform `INT4` and `INT8` GEMM operations on TensorCore. Further, for KV cache compression, we implement online quantization and packing for memory efficiency. For efficient implementation of online hadamard transform involved in activation quantization in `down_proj` layer, we use the fast hadamard transform library (fas, 2023).

**Prefill speedup** On an NVIDIA RTX 3090 GPU, we achieve a $1.61\times$ to $3.03\times$ speedup with ResQ over the 16-bit baseline for a single decoder block across various language models (Figure 5). Speedups are higher for larger models and shorter sequences. Compared to `INT4`, ResQ is only 14% slower on average, showing minimal overhead from mixed-precision and on-the-fly projections.

**Memory usage** We evaluate end to end memory usage on NVIDIA RTX 3090 (24GB) for different sequence lengths in Table 7. ResQ consumes $1.84\times$ - $3.08\times$ lower memory than the FP16 baseline. Notably, `Qwen2.5-14B` model leads to out of memory (OOM) error while ResQ is able to support its inference upto sequence length of 8192 tokens. Compared with the `INT4` baseline QuaRot, the memory used by ResQ is 4-11% higher.

**Multi GPU inference** We evaluate end-to-end batched inference latency on a GPU server with 3 NVIDIA A100 (82 GB) GPUs running `Meta-Llama-3-70B`. ResQ's weight, activation, and KV cache quantization enable the 70B model to fit on a single GPU, while FP16 requires model parallelism across all three GPUs. This allows ResQ to support data-parallel inference, unlike FP16. In Table 8, we show time to first token (i.e. end to end prefill latency) at different sequence lengths and batch sizes. Compared to FP16 baseline, ResQ achieves upto $4.98\times$ improvement in end to end latency under batched inference setting. This improvement stems from two factors, first is computational complexity reduction achieved in ResQ due to weight and activation quan-

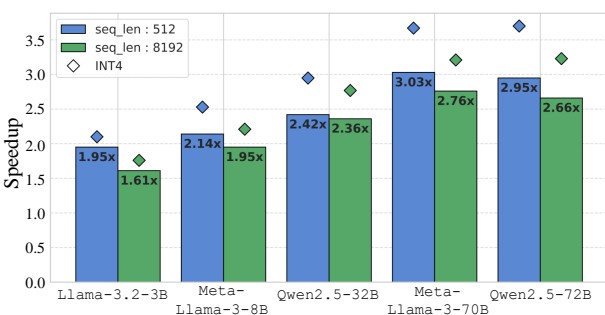

Figure 5: Speedup of ResQ and INT4 kernel on single decoder block on NVIDIA RTX 3090 over 16-bit floating point baseline for batch size of 1.

Table 7: Memory usage (in GB) for different sequence lengths on NVIDIA RTX 3090.

| Model | seq len | FP16 | QuaRot (Compression) | ResQ (Compression) |
|---|---|---|---|---|
| Meta-Llama-3-8B | 8192 | 21.9 | 11.4 (1.92×) | 11.9 (1.84×) |
| | 2048 | 16.7 | 6.8 (2.45×) | 7.2 (2.31×) |
| | 512 | 15.4 | 5.6 (2.75×) | 6.1 (2.54×) |
| Llama-2-7B-hf | 8192 | 18.1 | 6.2 (2.91×) | 6.8 (2.66×) |
| | 2048 | 13.9 | 4.2 (3.30×) | 4.7 (2.95×) |
| | 512 | 12.9 | 3.7 (3.48×) | 4.2 (3.08×) |
| Qwen2.5-14B | 8192 | OOM | 19.5 | 21.3 |
| | 2048 | OOM | 14.0 | 14.9 |
| | 512 | OOM | 12.6 | 13.5 |

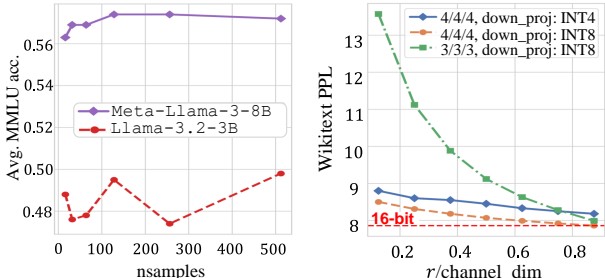

Figure 6: Ablation study on changing rank of high precision subspace for Llama-3.2-3B (left) and changing number of calibration samples (right).

tization, and, second is the memory compression achieved in ResQ due to weight and kv cache quantization which enables serving the 70B parameter model on a single GPU device.

### 5.4. Ablation Studies

**Projection bases** We evaluate the impact of different projections employed in ResQ by removing them and evaluating performance in Table 6. We see that removing $U_D$ or $U_A$ has a catastrophic impact on perplexity highlighting their importance. $U_B$ and $U_C$ which aid in quantization of KV cache have less severe impact when removed independently.

Table 8: Time to first token (in ms) for Meta-Llama-3-70B on GPU server with 3 NVIDIA A100 GPUs.

| batch size | seq len | FP16 | ResQ | Improv. over FP16 |
|---|---|---|---|---|
| 3 | 10240 | 20783 | 4242 | 4.90× |
| 3 | 8192 | 16373 | 3361 | 4.87× |
| 3 | 4096 | 7871 | 1609 | 4.89× |
| 3 | 2048 | 3888 | 806 | 4.82× |
| 6 | 2048 | 7733 | 1560 | 4.96× |
| 9 | 2048 | 11493 | 2309 | 4.98× |

But removing both of them leads to a non trivial increase in perplexity (particularly for Meta-Llama-3-8B and Llama-3.2-3B which employ grouped query attention).

**Rank of high-precision subspace** ResQ allows for seamless trade-off between accuracy and performance by modulating the rank $r$ of high precision subspace (Figure 6-left). Increasing the rank improves perplexity albeit at the cost of increased computations in high precision.

**Calibration dataset size** We change number of Wikitext calibration samples used to obtain projections and evaluate performance in Figure 6-right. For Meta-Llama-3-8B, MMLU accuracy increases with increasing samples and saturates beyond 128 samples. For Llama-3.2-3B, the trend is unclear with 512 samples achieving best performance.

**Calibration dataset** We evaluate the sensitivity of ResQ's projections to the calibration dataset. While the random rotation matrix $R$ is data-independent, the PCA-based projection matrix $P$ depends on the data. We obtain $P$ using samples from Alpaca (Taori et al., 2023), PTB (Marcus et al., 1993), and C4 (Raffel et al., 2020), and Table 16 in Appendix I shows minimal performance variation, demonstrating the robustness of ResQ's calibration.

## 6. Conclusion

We introduce *ResQ*, a novel mixed-precision, accelerator-friendly PTQ technique toward 4-bit quantization of large language models. ResQ projects weight, activation, and KV cache tensors to subspaces spanned by principal components, quantizing a low-rank ($1/8$ of hidden dimension) high-variance subspace to 8-bit and the rest to 4-bit. ResQ outperforms both uniform- and mixed-precision quantization methods. We demonstrate the effectiveness of ResQ across a variety of tasks—including language modeling, language understanding, common-sense reasoning, language generation and multi modal understanding—using the Llama and Qwen models. Compared to SpinQuant, the strongest baseline, ResQ achieves up to 33% lower perplexity on the WikiText dataset without requiring any additional training and offers up to $5\times$ speedup over the 16-bit baseline.

## Impact Statement

ResQ is a significant step forward towards efficiently serving LLMs in resource-constrained, on-device scenarios, potentially expanding the application space for these models. Although our approach aims to make LLMs more accessible and widely used, it does not address the potential risks of misuse for malicious purposes. To mitigate these risks, a strong commitment to user data protection, clear ethical guidelines, and transparency mechanisms is essential.

## Acknowledgements

The authors would like to thank Wanzin Yazar and Tristan Webb for infrastructure and technical assistance and Zifei Xu and Sakshi Choudhary for helpful discussions. The authors also thank Amogh Joshi for providing access to personal NVIDIA RTX 3090. This work was supported by the Center for the Co-Design of Cognitive Systems (CO-COSYS), a DARPA sponsored JUMP center of Semiconductor Research Corporation (SRC), Intel, SRC AIHW Program.

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

## A. Proof of Theorem 4.2

We begin the proof by introducing the following lemma.

**Lemma A.1.** *For any tensor $\boldsymbol{R}$ quantized following the quantization described in equation 1, assuming the values of $\boldsymbol{R}$ follows a normal distribution, we have*

$$\mathbb{E}\|\boldsymbol{R} - Q(\boldsymbol{R})\|_F \leq \frac{\sqrt{\pi \log\left[size(\boldsymbol{R})\right]}}{2^{n-1} - 1} \mathbb{E}\|\boldsymbol{R}\|_F \tag{9}$$

*where size($\boldsymbol{R}$) denotes the number of elements in $\boldsymbol{R}$.*

Proof of lemma A.1 can be found in (Li et al., 2025). From this lemma we obtain that the quantization error $\|\boldsymbol{R} - Q(\boldsymbol{R})\|_F$ is bounded by the magnitude of the tensor quantized $\|\boldsymbol{R}\|_F$. Now for our use case of mixed precision quantization where the low-precision component is quantized to $L$ bits and high precision component is quantized to $H$ bits, we write the quantization error again below,

$$\mathbb{E}\|\boldsymbol{X} - \boldsymbol{X}_q\|_F = \mathbb{E}\|\boldsymbol{X}\boldsymbol{U}_l - Q_L(\boldsymbol{X}\boldsymbol{U}_l)\|_F \\ + \mathbb{E}\|\boldsymbol{X}\boldsymbol{U}_h - Q_H(\boldsymbol{X}\boldsymbol{U}_h)\|_F. \tag{10}$$

The random rotation matrices $\boldsymbol{R}$ ensure that $\boldsymbol{X}\boldsymbol{U}_l$ and $\boldsymbol{X}\boldsymbol{U}_h$ are normally distributed by Lemma 4.1. Applying Lemma A.1 to the quantization error in equation 10, we get,

$$\begin{aligned} \|\boldsymbol{X} - \boldsymbol{X}_q\|_F &\leq \frac{\sqrt{\log(\text{size}(\boldsymbol{X}\boldsymbol{U}_l))\pi}}{2^{L-1} - 1} \mathbb{E}\|\boldsymbol{X}\boldsymbol{U}_l\|_F \\ &+ \frac{\sqrt{\log(\text{size}(\boldsymbol{X}\boldsymbol{U}_h))\pi}}{2^{H-1} - 1} \mathbb{E}\|\boldsymbol{X}\boldsymbol{U}_h\|_F \\ &= \frac{\sqrt{\log(\text{size}(\boldsymbol{X}\boldsymbol{P}_l))\pi}}{2^{L-1} - 1} \mathbb{E}\|\boldsymbol{X}\boldsymbol{P}_l\|_F \\ &+ \frac{\sqrt{\log(\text{size}(\boldsymbol{X}\boldsymbol{P}_h))\pi}}{2^{H-1} - 1} \mathbb{E}\|\boldsymbol{X}\boldsymbol{P}_h\|_F \\ &= \frac{\sqrt{\log(\text{size}(\boldsymbol{X}\boldsymbol{P}_l))\pi}}{2^{L-1} - 1} \mathbb{E}\|\text{tr}(\boldsymbol{X}\boldsymbol{P}_l\boldsymbol{P}_l^\top \boldsymbol{X}^\top)\|_F \\ &+ \frac{\sqrt{\log(\text{size}(\boldsymbol{X}\boldsymbol{P}_h)\pi}}{2^{H-1} - 1} \mathbb{E}\|\text{tr}(\boldsymbol{X}\boldsymbol{P}_h\boldsymbol{P}_h^\top \boldsymbol{X}^\top)\|_F \end{aligned} \tag{11}$$

We know $\text{size}(\boldsymbol{X}\boldsymbol{P}_l) = d - r$ and $\text{size}(\boldsymbol{X}\boldsymbol{P}_h) = r$ since $r$ components are in high precision. With $\boldsymbol{P}_l\boldsymbol{P}_L^\top + \boldsymbol{P}_h\boldsymbol{P}_h^\top = \boldsymbol{I}$, we have

$$\begin{aligned} \|\boldsymbol{X} - \boldsymbol{X}_q\|_F &\leq \frac{\sqrt{\log(\text{d-r})\pi}}{2^{L-1} - 1} (\mathbb{E}\|\boldsymbol{X}\|_F - \mathbb{E}\|\boldsymbol{X}\boldsymbol{P}_h\|_F) \\ &+ \frac{\sqrt{\log(\text{r})\pi}}{2^{H-1} - 1} \mathbb{E}\|\boldsymbol{X}\boldsymbol{P}_h\|_F \\ &= \frac{\sqrt{\log(\text{d-r})\pi}}{2^{L-1} - 1} \mathbb{E}\|\boldsymbol{X}\|_F \\ &- (\frac{\sqrt{\log(\text{d-r})\pi}}{2^{L-1} - 1} - \frac{\sqrt{\log(\text{r})\pi}}{2^{H-1} - 1}) \mathbb{E}\|\boldsymbol{X}\boldsymbol{P}_h\|_F \end{aligned} \tag{12}$$

Since $\frac{\sqrt{\log(\text{d-r})\pi}}{2^{L-1}-1} - \frac{\sqrt{\log(\text{r})\pi}}{2^{H-1}-1} > 0$ the quantization error is reduced by maximizing $\|\boldsymbol{X}\boldsymbol{P}_h\|_F$

## B. Distribution of activations

The distribution of activations after projection by $\boldsymbol{U}$ is shown in Figure 7. The formulation of $\boldsymbol{U}$ ensures that the final $r$ channels in the activation map comprise of coefficients along bases with maximum activation variance. Consequently, keep those channels in high precision minimizes quantization error. The remaining channels are more amenable to quantization due to the application of random rotations which suppress outlier values.

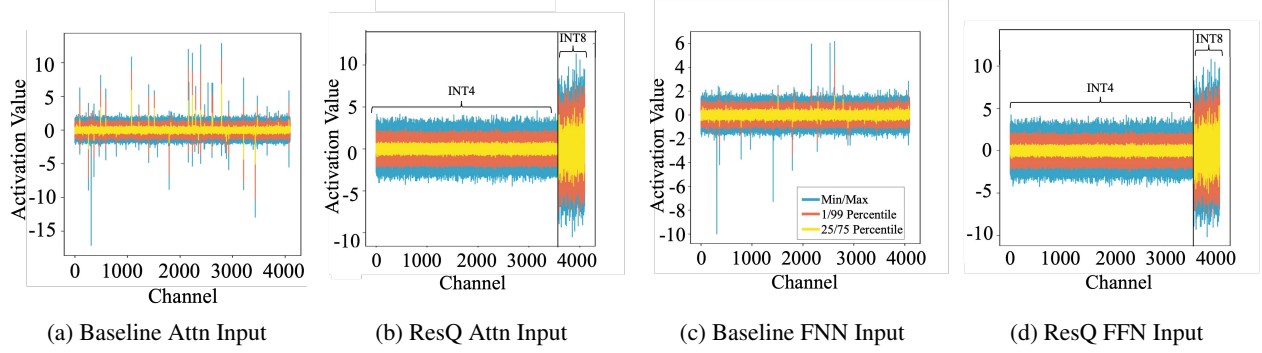

| | | | |
|---|---|---|---|
| (a) Baseline Attn Input | (b) ResQ Attn Input | (c) Baseline FNN Input | (d) ResQ FFN Input |

Figure 7: Input activation distributions of attention and FFN layers, for baseline (a and c) and ResQ (b and d).

Table 9: Time taken (in NVIDIA A100 GPU hours) to quantize the model. All approaches use GPTQ (Frantar et al., 2023) for weight quantization. SpinQuant uses 4 GPUs to optimize rotation matrices.

| | QuaRot | ResQ | SpinQuant |
|---|---|---|---|
| Llama-3.2-1B | 4m | 7m | 13m |
| Llama-3.2-3B | 8m | 16m | 38m |
| Meta-Llama-3-8B | 17m | 35m | 1h41m |
| Llama-2-7b-hf | 15m | 33m | 1h37m |
| Llama-2-13b-hf | 23m | 1h | 3h42m |

## C. Additional implementation details

In this work, obtaining the projection matrices and quantization of weights for **all the models** is performed on a single NVIDIA A100 80GB GPUs. Time taken by ResQ compared with other approaches is shown in Table 9. Evaluation on various benchmarks for all the models is also done on a single NVIDIA A100 GPU with the sole exception of `Meta-Llama-3-70b` which requires 4 GPUs for evaluation. We use `lm_evaluation_harness` version 0.4.5 (Gao et al., 2024) and LongBench (Bai et al., 2024) for all the evaluation tasks. For Arc-c/e, Hellaswag, OpenBook QA, PIQA tasks we report `acc_norm` while for BoolQ, SIQA and Winogrande we report `acc`.

For calibration data, we use 512 randomly choses samples for Wikitext to obtain the projection matrices. While for GPTQ we use 128 randomly choses samples from Wiktiext following the original work Frantar et al. 2023.

The KV cache, as well as the weights and activations of all Linear layers (except `mlp.down_proj`), are quantized to 4-bit precision, with $\frac{1}{8}$ of channels retained in 8-bit precision. While, the weights and activations within `down_proj` are uniformly quantized to 4-bit precision. Following Ashkboos et al. 2024c and Liu et al. 2025, we keep query vector in 16-bit.

## D. Complete results of main result tables

Detailed results of Table 1 in the main paper, including more models and task-by-task performance, are shown in Tables 10 (Llama families) and 11 (Qwen2.5 family). As expected, ResQ achieves superior performance to baselines across the series of common sense reasoning and MMLU tasks.

## E. Additional quantization results: W/A/KV = 4/8/4 bits of precision

This section presents additional comparisons between baselines and ResQ for the Llama family when quantized to **W/A/KV = 4/8/4** bits of precision. Across various MMLU tasks and perplexity evaluations on WikiText, ResQ consistently outperforms all baselines. For 0-shot common sense reasoning tasks, except for `Meta-Llama-3-8B`, ResQ achieves the best average performance. In the case of `Meta-Llama-3-8B`, ResQ is the second-best method, with QuaRot performing marginally better by less than 0.2%.

Table 10: Comparison of perplexity on Wikitext, accuracy on eight 0-shot common sense reasoning tasks including ARC-challenge, ARC-easy, BoolQ, HellaSwag, Openbook QA, PIQA, SIQA, and WinoGrande, and 0-shot massive multitask language understanding tasks across four subjects: STEM, Humanities, Social Sciences, and MMLU-other, for the Llama 2, Llama 3 and Llama 3.2 families. Results of all techniques were obtained using their official codebase. Our work ResQ and QUIK (Ashkboos et al., 2024b) keep 1/8 channels in 8-bit and remaining in 4-bit for W/A/KV = 4.5-bit. Other baselines are uniformly quantized to W/A/KV = 4-bit. All techniques except RTN use GPTQ (Frantar et al., 2023). (↓): lower is better, (↑): higher is better.

**Llama 2 family**

| Model | Method | Perplexity Wiki (↓) | ARC-c (↑) | ARC-e (↑) | BoolQ (↑) | HellaS (↑) | OBQA (↑) | PIQA (↑) | SIQA (↑) | WinoG (↑) | Avg. (↑) | humanities (↑) | Other (↑) | SocialS (↑) | STEM (↑) | Avg. (↑) |
|---|---|---|---|---|---|---|---|---|---|---|---|---|---|---|---|---|
| Llama-2-7b-hf | 16-bit | 5.5 | 46.3 | 74.6 | 77.8 | 75.9 | 44.2 | 79.2 | 46.1 | 69.1 | 64.1 | 38.9 | 45.9 | 46.0 | 33.4 | 41.1 |
| | RTN | 1766.2 | 26.3 | 27.8 | 54.8 | 29.4 | 25.8 | 51.0 | 35.0 | 48.7 | 37.4 | 24.5 | 24.7 | 22.9 | 22.2 | 23.6 |
| | GPTQ | 9600.0 | 24.8 | 31.4 | 55.4 | 30.6 | 25.6 | 55.8 | 34.2 | 53.3 | 38.9 | 24.7 | 24.5 | 22.7 | 23.2 | 23.8 |
| | SmoothQuant+ | 15.4 | 29.3 | 47.1 | 56.8 | 48.6 | 31.8 | 65.5 | 37.2 | 52.4 | 46.1 | 25.0 | 24.5 | 24.1 | 23.4 | 24.2 |
| | QUIK | 7.5 | 39.8 | 63.7 | 68.9 | 68.3 | 37.8 | 72.9 | 42.1 | 62.4 | 57.0 | 26.9 | 29.6 | 28.8 | 25.8 | 27.8 |
| | QuaRot | 6.1 | 41.5 | 71.4 | 73.2 | 73.2 | 40.6 | 76.9 | 43.6 | 65.6 | 60.7 | 31.2 | 35.1 | 34.6 | 28.2 | 32.3 |
| | SpinQuant | 6.0 | 43.6 | 71.3 | 73.8 | 73.2 | 40.4 | 76.0 | **44.1** | 65.4 | 61.0 | 33.9 | 38.5 | 37.5 | 29.5 | 34.8 |
| | ResQ | **5.8** | **44.0** | **72.6** | **75.3** | **74.0** | **41.0** | **77.9** | 43.9 | **66.9** | **62.0** | **35.9** | **40.9** | **42.2** | **32.2** | **37.7** |
| Llama-2-13b-hf | 16-bit | 4.9 | 49.1 | 77.4 | 80.5 | 79.4 | 45.2 | 80.7 | 47.2 | 72.1 | 66.5 | 47.9 | 59.3 | 61.0 | 42.4 | 52.7 |
| | RTN | 3543.9 | 22.8 | 29.8 | 40.2 | 26.6 | 27.8 | 51.4 | 35.6 | 50.6 | 33.5 | 23.7 | 25.0 | 23.1 | 22.6 | 23.6 |
| | GPTQ | 3120.0 | 23.6 | 31.1 | 38.7 | 27.2 | 26.8 | 53.6 | 35.8 | 49.8 | 33.8 | 25.0 | 25.4 | 23.7 | 25.1 | 24.8 |
| | SmoothQuant+ | 11.2 | 34.5 | 55.6 | 62.9 | 62.5 | 32.4 | 70.1 | 38.7 | 55.6 | 51.0 | 25.7 | 26.1 | 27.3 | 27.3 | 26.6 |
| | QUIK | 6.8 | 43.7 | 68.0 | 71.3 | 73.3 | 40.0 | 75.7 | 45.1 | 64.6 | 60.2 | 34.7 | 40.6 | 39.8 | 31.8 | 36.7 |
| | QuaRot | 5.4 | 46.9 | 74.9 | 76.6 | 75.8 | 42.6 | 79.1 | 45.5 | 69.0 | 63.8 | 43.8 | 53.6 | 54.0 | 39.4 | 47.7 |
| | SpinQuant | 5.2 | 49.0 | **76.3** | 78.2 | 77.1 | 42.8 | **79.3** | 46.3 | 69.5 | 64.8 | 43.5 | 53.1 | 55.4 | 39.1 | 47.8 |
| | ResQ | **5.1** | **49.1** | 76.1 | **79.7** | **77.9** | **43.6** | 79.1 | **46.6** | **69.9** | **65.2** | **45.3** | **56.0** | **58.0** | **41.0** | **50.1** |

**Llama 3 family**

| Model | Method | Perplexity Wiki (↓) | ARC-c (↑) | ARC-e (↑) | BoolQ (↑) | HellaS (↑) | OBQA (↑) | PIQA (↑) | SIQA (↑) | WinoG (↑) | Avg. (↑) | humanities (↑) | Other (↑) | SocialS (↑) | STEM (↑) | Avg. (↑) |
|---|---|---|---|---|---|---|---|---|---|---|---|---|---|---|---|---|
| Meta-Llama-3-8B | 16-bit | 6.1 | 53.2 | 77.1 | 81.1 | 79.2 | 44.8 | 80.9 | 47.0 | 73.4 | 67.1 | 55.0 | 70.6 | 73.2 | 53.7 | 63.1 |
| | RTN | 218.9 | 25.3 | 34.9 | 44.2 | 38.3 | 27.8 | 56.5 | 36.8 | 50.8 | 39.3 | 24.7 | 25.1 | 23.3 | 21.4 | 23.6 |
| | GPTQ | 166.3 | 24.7 | 37.7 | 44.3 | 36.8 | 27.0 | 57.6 | 36.4 | 53.8 | 39.8 | 24.7 | 23.9 | 22.8 | 21.8 | 23.3 |
| | SmoothQuant+ | 78.2 | 27.5 | 42.0 | 50.7 | 44.9 | 28.8 | 59.0 | 35.9 | 50.9 | 42.5 | 25.4 | 25.5 | 24.5 | 23.4 | 24.7 |
| | QUIK | 14.2 | 33.6 | 56.4 | 60.5 | 61.5 | 33.2 | 68.7 | 39.9 | 59.0 | 51.6 | 30.0 | 34.0 | 34.8 | 32.1 | 32.7 |
| | QuaRot | 7.8 | 45.1 | 70.4 | 73.8 | 74.7 | 42.6 | 76.6 | 45.1 | 68.5 | 62.1 | 47.8 | 59.1 | 61.4 | 44.3 | 53.2 |
| | SpinQuant | 7.4 | 48.0 | **75.4** | **75.8** | 75.4 | **43.8** | 77.5 | 45.0 | 69.2 | 63.8 | 49.8 | 63.3 | 65.0 | 46.8 | 56.2 |
| | ResQ | **7.1** | **49.2** | 75.0 | 72.5 | **76.5** | 43.0 | **78.3** | **45.8** | **71.0** | **63.9** | **50.6** | **64.4** | **65.8** | **48.1** | **57.2** |
| Meta-Llama-3-70B | 16-bit | 2.9 | 64.2 | 85.9 | 85.3 | 84.9 | 48.6 | 84.4 | 50.8 | 80.6 | 73.1 | 67.6 | 81.5 | 86.8 | 68.4 | 76.1 |
| | RTN | 452.7 | 32.6 | 50.3 | 54.2 | 41.3 | 31.6 | 64.8 | 35.9 | 53.2 | 45.5 | 24.5 | 23.8 | 22.3 | 22.1 | 23.2 |
| | GPTQ | 11655.0 | 25.9 | 26.0 | 37.9 | 26.2 | 28.6 | 50.4 | 34.3 | 49.9 | 34.9 | 27.1 | 24.3 | 24.0 | 26.5 | 25.5 |
| | SmoothQuant+ | - | - | - | - | - | - | - | - | - | - | - | - | - | - | - |
| | QUIK | 8.0 | 44.5 | 68.9 | 60.7 | 75.0 | 36.4 | 76.1 | 43.2 | 60.4 | 58.2 | 46.6 | 56.4 | 58.0 | 43.6 | 51.1 |
| | QuaRot | 5.7 | 53.7 | 74.5 | 81.6 | 81.1 | **46.6** | 81.0 | 46.8 | 75.2 | 67.6 | 55.7 | 72.5 | 75.8 | 57.3 | 65.3 |
| | SpinQuant | 6.2 | 52.0 | 77.3 | 81.7 | 75.6 | 43.8 | 78.8 | 43.4 | 72.8 | 65.7 | 50.7 | 67.0 | 68.1 | 51.9 | 59.4 |
| | ResQ | **4.1** | **61.4** | **84.3** | **83.9** | **83.5** | 46.0 | **83.1** | **48.6** | **78.3** | **71.1** | **64.9** | **79.9** | **84.9** | **66.1** | **74.0** |

**Llama 3.2 family**

| Model | Method | Perplexity Wiki (↓) | ARC-c (↑) | ARC-e (↑) | BoolQ (↑) | HellaS (↑) | OBQA (↑) | PIQA (↑) | SIQA (↑) | WinoG (↑) | Avg. (↑) | humanities (↑) | Other (↑) | SocialS (↑) | STEM (↑) | Avg. (↑) |
|---|---|---|---|---|---|---|---|---|---|---|---|---|---|---|---|---|
| Llama-3.2-1B | 16-bit | 9.8 | 36.5 | 60.6 | 63.4 | 63.6 | 37.4 | 74.5 | 42.8 | 60.1 | 54.9 | 34.8 | 41.1 | 39.9 | 32.0 | 36.9 |
| | RTN | 329.1 | 22.4 | 29.9 | 53.4 | 31.4 | 29.4 | 54.8 | 34.9 | 48.5 | 38.1 | 24.8 | 25.2 | 22.4 | 22.7 | 23.8 |
| | GPTQ | 108.9 | 24.7 | 32.7 | 52.3 | 30.7 | 23.6 | 54.3 | 34.4 | 51.1 | 38.0 | 24.7 | 25.1 | 25.5 | 24.5 | 24.9 |
| | SmoothQuant+ | 228.9 | 23.3 | 30.1 | 52.9 | 31.3 | 26.6 | 54.2 | 34.5 | 51.2 | 38.0 | 23.9 | 24.1 | 25.0 | 23.5 | 24.1 |
| | QUIK | 21.8 | 27.4 | 46.0 | 55.0 | 46.0 | 26.4 | 62.4 | 38.6 | 52.6 | 44.3 | 25.6 | 25.6 | 24.6 | 24.5 | 25.1 |
| | QuaRot | 14.3 | 30.0 | 51.4 | 59.1 | 54.0 | 34.2 | 66.7 | 39.6 | 57.1 | 49.0 | 25.4 | 26.9 | 25.4 | 24.4 | 25.5 |
| | SpinQuant | 13.6 | 32.3 | 51.8 | **59.3** | 55.4 | 30.4 | 67.7 | 38.6 | 54.7 | 48.8 | 25.4 | 27.6 | 24.2 | 25.3 | 25.6 |
| | ResQ | **12.4** | **34.0** | **54.2** | 57.0 | **57.3** | 31.2 | **69.4** | **41.0** | 56.8 | **50.1** | **28.3** | **30.5** | **31.3** | **27.6** | **29.4** |
| Llama-3.2-3B | 16-bit | 7.8 | 46.2 | 71.7 | 73.1 | 73.7 | 37.4 | 77.4 | 47.2 | 69.1 | 62.7 | 48.9 | 62.9 | 62.3 | 45.2 | 54.8 |
| | RTN | 268.8 | 23.5 | 35.4 | 46.2 | 35.6 | 28.2 | 56.3 | 33.6 | 50.6 | 38.7 | 25.1 | 25.6 | 27.0 | 24.9 | 25.7 |
| | GPTQ | 178.3 | 27.0 | 27.0 | 48.8 | 44.4 | 27.8 | 59.1 | 37.1 | 51.5 | 40.3 | 24.9 | 24.5 | 25.7 | 24.0 | 24.8 |
| | SmoothQuant+ | 96.1 | 25.3 | 33.1 | 47.8 | 37.7 | 25.2 | 56.2 | 35.8 | 50.9 | 39.0 | 25.4 | 26.6 | 26.4 | 25.3 | 25.9 |
| | QUIK | 15.8 | 32.9 | 50.1 | 52.6 | 59.1 | 33.2 | 68.7 | 40.3 | 53.0 | 48.8 | 29.0 | 33.2 | 31.9 | 30.3 | 31.1 |
| | QuaRot | 10.1 | 38.6 | 59.0 | 65.9 | 66.5 | 35.8 | 74.4 | 43.1 | **65.2** | 56.1 | 38.5 | 47.3 | 46.7 | 35.3 | 42.0 |
| | SpinQuant | 9.2 | 38.9 | 64.8 | 68.0 | 69.1 | **39.4** | 74.9 | 45.1 | 62.9 | 57.9 | 37.0 | 49.4 | 50.5 | 39.9 | 44.2 |
| | ResQ | **8.8** | **43.1** | **65.6** | **68.8** | **70.5** | 38.4 | **75.1** | **45.6** | 64.8 | **59.0** | **44.7** | **57.0** | **56.5** | **41.0** | **49.8** |

# F. Complete results of the MMMU benchmark

This section presents task-by-task results for the MMMU benchmark across six subjects—Art & Design, Business, Science, Health & Medicine, Humanities & Social Science, and Tech & Engineering—for the Qwen2 VL family when quantized to **W/A/KV = 4/4/4** bits and **W/A/KV = 4/8/4** bits of precision. On average, ResQ consistently outperforms all baselines across different models. Notably, the advantage of ResQ becomes more pronounced with larger models. For instance, for Qwen2-VL-7B-Instruct at **W/A/KV = 4/8/4** bits of precision, ResQ achieves an average accuracy score of 48.8, significantly outperforming the next-best method, QUIK, which scores 26.4, representing an ∼ 85% relative improvement.

Table 11: Comparison of perplexity score on Wikitext, accuracy on eight 0-shot common sense reasoning tasks including ARC-challenge, ARC-easy, BoolQ, HellaSwag, Openbook QA, PIQA, SIQA, and WinoGrande, and 0-shot massive multitask language understanding tasks across four subjects: STEM, Humanities, Social Sciences, and MMLU-other, for the Qwen2.5 family. Results of all techniques were obtained using their official codebase. Our work ResQ and QUIK (Ashkboos et al., 2024b) keep $1/8$ channels in 8-bit and remaining in 4-bit for W/A/KV = 4.5-bit. Other baselines are uniformly quantized to W/A/KV = 4-bit. All techniques except RTN use GPTQ (Frantar et al., 2023). (↓): lower is better, (↑): higher is better, ∗: In cases where Hadamard matrix does not exist at the MLP dimension, random orthogonal rotation is used instead.

| | | Qwen2.5 family | | | | | | | | | | | | | | |
|---|---|---|---|---|---|---|---|---|---|---|---|---|---|---|---|---|
| | | Perplexity | 0-shot common sense reasoning tasks | | | | | | | | | 0-shot MMLU tasks | | | | |
| Model | Method | Wiki | ARC-c | ARC-e | BoolQ | HellaS | OBQA | PIQA | SIQA | WinoG | Avg. | humanities | Other | SocialS | STEM | Avg. |
| | | (↓) | (↑) | (↑) | (↑) | (↑) | (↑) | (↑) | (↑) | (↑) | (↑) | (↑) | (↑) | (↑) | (↑) | (↑) |
| Qwen2.5-0.5B | 16-bit | 13.1 | 31.9 | 58.4 | 62.1 | 52.1 | 35.0 | 69.7 | 44.3 | 57.1 | 51.3 | 42.2 | 53.2 | 55.5 | 41.5 | 48.1 |
| | RTN | 23204.3 | 26.2 | 27.0 | 39.3 | 26.0 | 24.0 | 50.7 | 34.5 | 51.5 | 34.9 | 24.8 | 24.0 | 22.8 | 24.3 | 23.9 |
| | GPTQ | 16302.3 | 23.7 | 26.9 | 39.0 | 26.5 | 26.4 | 50.2 | 33.4 | 49.6 | 34.5 | 24.1 | 24.8 | 23.5 | 23.0 | 23.9 |
| | SmoothQuant+ | 10053.9 | 25.9 | 26.3 | 39.9 | 27.2 | 25.4 | 47.1 | 35.9 | 49.6 | 34.7 | 24.5 | 24.7 | 21.5 | 22.1 | 23.2 |
| | QUIK | 38.6 | 24.5 | 38.6 | 48.0 | 36.9 | 28.4 | 58.1 | **36.4** | **51.9** | 40.4 | **26.3** | 25.9 | 23.6 | 24.2 | 25.0 |
| | QuaRot∗ | 219.9 | 25.4 | 36.6 | 45.0 | 28.9 | **28.6** | 54.1 | 32.9 | 51.7 | 37.9 | 24.4 | 24.0 | 23.0 | 23.5 | 23.7 |
| | ResQ | 29.6 | **27.1** | **44.2** | **53.2** | **38.8** | 28.0 | **61.9** | 34.4 | 51.3 | **42.4** | 26.1 | **27.5** | **25.3** | **26.0** | **26.2** |
| Qwen2.5-1.5B | 16-bit | 9.3 | 45.1 | 72.1 | 72.9 | 67.7 | 40.2 | 76.3 | 48.8 | 63.7 | 60.8 | 53.5 | 65.5 | 70.6 | 52.8 | 60.6 |
| | RTN | 14518.9 | 23.1 | 27.2 | 43.9 | 26.8 | 25.6 | 51.3 | 33.4 | 52.5 | 35.5 | 23.8 | 24.5 | 23.8 | 22.7 | 23.7 |
| | GPTQ | 25769.6 | 23.9 | 26.9 | 43.9 | 26.1 | 27.6 | 49.7 | 32.1 | 51.5 | 35.2 | 24.6 | 24.7 | 23.7 | 23.8 | 24.2 |
| | SmoothQuant+ | 31655.9 | 25.0 | 26.2 | 39.9 | 26.0 | 26.0 | 50.8 | 32.1 | 49.0 | 34.4 | 25.5 | 24.4 | 22.7 | 22.4 | 23.8 |
| | QUIK | 6613.5 | 21.8 | 31.9 | 40.9 | 27.9 | 27.4 | 52.8 | 35.2 | 48.6 | 35.8 | 24.6 | 24.0 | 21.9 | 21.7 | 23.1 |
| | QuaRot | 6599.9 | 23.6 | 37.3 | 46.2 | 28.6 | 27.0 | 56.3 | 35.2 | 52.4 | 38.3 | 24.5 | 24.3 | 23.0 | 22.4 | 23.5 |
| | ResQ | 12.5 | **38.7** | **64.1** | **65.7** | **61.4** | **37.8** | **71.6** | **42.7** | **60.1** | **55.3** | **43.2** | **54.4** | **54.9** | **41.5** | **48.5** |
| Qwen2.5-3B | 16-bit | 8.0 | 47.4 | 73.0 | 77.5 | 73.6 | 42.0 | 78.7 | 49.9 | 68.4 | 63.8 | 56.6 | 71.0 | 76.3 | 60.6 | 66.1 |
| | RTN | 39033.0 | 25.6 | 25.8 | 41.7 | 26.3 | 27.4 | 49.5 | 33.1 | 51.4 | 35.1 | 24.5 | 24.4 | 22.8 | 21.9 | 23.4 |
| | GPTQ | 9977.8 | 26.0 | 26.7 | 41.5 | 26.7 | 28.2 | 51.5 | 31.9 | 48.3 | 35.1 | 24.3 | 23.8 | 22.8 | 21.8 | 23.2 |
| | SmoothQuant+ | 73306.7 | 25.4 | 24.5 | 41.0 | 26.4 | 29.8 | 48.4 | 32.4 | 50.4 | 34.8 | 25.6 | 24.7 | 23.1 | 22.4 | 23.9 |
| | QUIK | 15.5 | 36.1 | 55.4 | 61.4 | 57.2 | 36.2 | 67.1 | 40.8 | 55.3 | 51.2 | 36.4 | 42.8 | 42.4 | 36.1 | 39.4 |
| | QuaRot | 68.8 | 32.4 | 53.1 | 51.6 | 49.2 | 33.4 | 66.7 | 39.3 | 56.4 | 47.7 | 28.1 | 32.0 | 28.9 | 26.6 | 28.9 |
| | ResQ | 9.0 | **45.3** | **70.5** | **72.7** | **70.2** | **42.4** | **76.8** | **46.7** | **64.4** | **61.1** | **53.1** | **66.5** | **70.5** | **54.8** | **61.2** |
| Qwen2.5-7B | 16-bit | 6.8 | 51.2 | 77.6 | 84.7 | 78.9 | 47.2 | 80.0 | 54.8 | 73.2 | 68.4 | 62.6 | 76.7 | 82.6 | 70.1 | 73.0 |
| | RTN | 24382.1 | 24.5 | 26.3 | 37.8 | 26.0 | 29.0 | 51.0 | 34.1 | 50.1 | 34.9 | 24.9 | 24.3 | 23.4 | 24.9 | 24.4 |
| | GPTQ | 13593.7 | 25.2 | 25.6 | 37.8 | 26.3 | 28.2 | 52.4 | 34.4 | 48.9 | 34.8 | 24.4 | 24.3 | 22.8 | 22.6 | 23.5 |
| | SmoothQuant+ | 19088.7 | 26.3 | 25.2 | 39.8 | 26.4 | 27.6 | 52.7 | 33.5 | 52.0 | 35.4 | 25.1 | 25.4 | 22.6 | 24.1 | 24.3 |
| | QUIK | 260.3 | 29.5 | 42.4 | 51.7 | 36.3 | 28.2 | 59.6 | 34.5 | 49.6 | 41.5 | 24.3 | 26.9 | 23.1 | 23.8 | 24.6 |
| | QuaRot∗ | 4035.9 | 25.9 | 41.0 | 39.1 | 29.1 | 27.6 | 57.9 | 35.7 | 50.6 | 38.4 | 24.8 | 24.4 | 24.4 | 22.7 | 24.1 |
| | ResQ | 8.2 | **49.0** | **74.7** | **81.4** | **75.7** | **45.0** | **78.9** | **49.4** | **68.2** | **65.3** | **57.8** | **74.4** | **79.3** | **64.5** | **69.0** |
| Qwen2.5-14B | 16-bit | 5.3 | 58.8 | 79.4 | 85.4 | 82.9 | 45.4 | 81.9 | 55.3 | 75.8 | 70.6 | 69.9 | 81.9 | 86.2 | 76.5 | 78.6 |
| | RTN | 2715 | 21.6 | 32.7 | 51.5 | 29.6 | 25.8 | 52.6 | 33.2 | 51.7 | 37.3 | 25.3 | 23.2 | 26.0 | 25.3 | 24.9 |
| | GPTQ | 5100.3 | 23.8 | 29.1 | 47.7 | 30.1 | 27.6 | 51.3 | 34.6 | 51.2 | 34.8 | 25.1 | 24.7 | 25.1 | 24.3 | 24.8 |
| | SmoothQuant+ | 1375.7 | 27.0 | 26.3 | 38.0 | 26.8 | 29.2 | 51.6 | 32.4 | 49.3 | 35.1 | 25.9 | 24.5 | 22.2 | 22.2 | 23.7 |
| | QUIK | 10.5 | 45.0 | 67.1 | 64.7 | 68.9 | 37.6 | 74.8 | 43.9 | 59.3 | 57.6 | 48.9 | 61.1 | 64.7 | 51.5 | 56.6 |
| | QuaRot | 6.8 | 54.8 | 79.6 | 79.9 | 78.7 | 44.0 | 79.5 | 49.7 | 70.7 | 67.1 | 60.9 | 75.1 | 80.2 | 67.3 | 70.9 |
| | ResQ | 6.2 | **57.6** | **82.1** | **84.9** | **81.1** | **44.8** | **80.5** | **51.7** | 70.6 | **69.2** | **65.2** | **78.4** | **83.4** | **71.5** | **74.6** |
| Qwen2.5-32B | 16-bit | 5.0 | 55.7 | 78.0 | 87.4 | 84.1 | 44.4 | 82.3 | 56.4 | 75.2 | 70.4 | 73.1 | 83.6 | 89.6 | 81.2 | 81.9 |
| | RTN | 1847.4 | 24.3 | 35.3 | 51.4 | 31.9 | 27.0 | 52.8 | 34.1 | 51.4 | 38.5 | 24.5 | 25.1 | 25.3 | 24.3 | 24.8 |
| | GPTQ | 3891.1 | 25.4 | 35.4 | 48.5 | 31.8 | 27.0 | 53.8 | 35.8 | 50.5 | 38.5 | 25.9 | 24.8 | 23.6 | 24.0 | 24.6 |
| | SmoothQuant+ | - | - | - | - | - | - | - | - | - | - | - | - | - | - | - |
| | QUIK | 9.6 | 41.0 | 64.6 | 74.9 | 72.0 | 39.6 | 75.8 | 44.5 | 60.2 | 59.1 | 54.7 | 66.8 | 71.3 | 58.8 | 62.9 |
| | QuaRot | 6.1 | 54.5 | 76.1 | 85.1 | 81.5 | 44.2 | 80.1 | 51.3 | 70.4 | 67.9 | 68.5 | 80.0 | 86.0 | 76.0 | 77.6 |
| | ResQ | 5.6 | **55.1** | **78.4** | **86.0** | **82.5** | **45.4** | **81.1** | **53.9** | **74.0** | **69.5** | **70.3** | **82.3** | **87.9** | **78.9** | **79.8** |
| Qwen2.5-72B | 16-bit | 3.9 | 62.6 | 83.2 | 89.2 | 86.0 | 46.6 | 83.6 | 58.4 | 77.7 | 73.4 | 77.2 | 86.9 | 90.6 | 82.4 | 84.3 |
| | RTN | 45412.7 | 25.9 | 26.3 | 38.0 | 25.9 | 25.2 | 50.0 | 34.2 | 48.7 | 34.3 | 25.5 | 24.2 | 23.0 | 23.2 | 24.0 |
| | GPTQ | 37967.2 | 25.4 | 25.8 | 38.1 | 25.6 | 26.6 | 51.2 | 34.2 | 49.4 | 34.5 | 25.1 | 24.0 | 21.9 | 22.2 | 23.3 |
| | SmoothQuant+ | - | - | - | - | - | - | - | - | - | - | - | - | - | - | - |
| | QUIK | 8.3 | 45.1 | 68.1 | 77.2 | 77.2 | 39.0 | 77.4 | 45.6 | 65.6 | 61.9 | 60.2 | 74.3 | 77.5 | 65.3 | 69.3 |
| | QuaRot | 4.9 | 55.8 | **81.1** | 87.5 | 84.0 | 45.2 | 81.7 | 52.5 | 74.5 | 70.3 | 71.4 | 84.2 | 87.7 | 77.1 | 80.1 |
| | ResQ | 4.6 | **58.4** | 80.9 | **88.4** | **84.9** | **48.2** | **82.6** | **55.5** | **77.0** | **72.0** | **72.8** | **84.6** | **89.0** | **79.5** | **81.5** |

# G. Complete results of iso-bitwidth comparison

This section presents detailed results for Table 5 in the main paper for Qwen2.5 family of models. The task-by-task results are provided in Table 14. As expected, ResQ achieves superior performance. For Qwen2.5-72B, SpinQuant baseline is missing because of lack of big enough GPU cluster to enable training rotations.

# H. Complete results when training rotation matrix $R$

In Table 15, we provide results on Wikitext perplexity, task-by-task accuracy on 0-shot reasoning benchmarks and task-by-task accuracy on MMLU benchmark when training rotation matrix $R$ in ResQ. The rotation matrix is trained using Cayley SGD (Li et al., 2020) for 100 training steps at batch size 8 and learning rate of 1.5. The training data involves samples

Table 12: Accuracy on eight 0-shot common sense reasoning tasks including ARC-challenge, ARC-easy, BoolQ, HellaSwag, Openbook QA, PIQA, SIQA, and WinoGrande and 0-shot massive multitask language understanding tasks across four subjects: STEM, Humanities, Social Sciences, and MMLU-other, for the Llama 2, Llama 3 and Llama 3.2 families when quantized to **W/A/KV = 4/8/4** bits. Results of all techniques were obtained using their official codebase. Our work ResQ and QUIK (Ashkboos et al., 2024b) keep $1/8$ of channels in 8-bit. All techniques except RTN use GPTQ (Frantar et al., 2023). ($\downarrow$): lower is better, ($\uparrow$): higher is better.

| | | Perplexity | 0-shot common sense reasoning tasks | | | | | | | | | 0-shot MMLU tasks | | | | |
|---|---|---|---|---|---|---|---|---|---|---|---|---|---|---|---|---|
| Model | Method | Wiki ($\downarrow$) | ARC-c ($\uparrow$) | ARC-e ($\uparrow$) | BoolQ ($\uparrow$) | HellaS ($\uparrow$) | OBQA ($\uparrow$) | PIQA ($\uparrow$) | SIQA ($\uparrow$) | WinoG ($\uparrow$) | Avg. ($\uparrow$) | humanities ($\uparrow$) | Other ($\uparrow$) | SocialS ($\uparrow$) | STEM ($\uparrow$) | Avg. ($\uparrow$) |
| | | | | | | | Llama 2 family | | | | | | | | | |
| Llama-2-7b-hf | 16-bit | 5.5 | 46.3 | 74.6 | 77.8 | 75.9 | 44.2 | 79.2 | 46.1 | 69.1 | 64.1 | 38.9 | 45.9 | 46.0 | 33.4 | 41.1 |
| | RTN | 7.2 | 41.5 | 65.9 | 71.9 | 71.8 | 39.4 | 76.5 | 42.7 | 65.7 | 59.4 | 27.6 | 28.4 | 31.6 | 29.0 | 29.2 |
| | GPTQ | 11.8 | 42.5 | 71.3 | 69.9 | 73.6 | 43.2 | 77.4 | 44.9 | 68.9 | 61.5 | 28.0 | 32.3 | 32.1 | 28.4 | 30.2 |
| | SmoothQuant+ | 6.8 | 41.9 | 69.1 | 70.7 | 72.9 | 40.2 | 77.1 | 32.7 | 66.9 | 58.9 | 28.1 | 30.0 | 28.6 | 27.1 | 28.5 |
| | QUIK | 5.7 | 43.9 | 73.4 | 77.3 | 74.2 | 44.6 | 78.2 | 44.3 | 68.9 | 63.1 | 35.8 | 39.3 | 40.1 | 30.4 | 36.4 |
| | QuaRot | 5.7 | 43.6 | 73.6 | 75.4 | 74.8 | 42.6 | 77.6 | 45.1 | 67.9 | 62.6 | 36.3 | 43.3 | 41.5 | 31.3 | 38.1 |
| | SpinQuant | 5.7 | 43.5 | 73.3 | 75.4 | 74.8 | 42.6 | 77.5 | 45.0 | 68.4 | 62.6 | 37.0 | 42.0 | 43.4 | 31.8 | 38.5 |
| | ResQ | 5.6 | 46.3 | 74.5 | 77.1 | 75.0 | 44.2 | 78.9 | 45.6 | 68.8 | 63.6 | 39.1 | 45.9 | 47.9 | 34.9 | 42.0 |
| Llama-2-13b-hf | 16-bit | 4.9 | 49.1 | 77.4 | 80.5 | 79.4 | 45.2 | 80.7 | 47.2 | 72.1 | 66.5 | 47.9 | 59.3 | 61.0 | 42.4 | 52.7 |
| | RTN | 6.9 | 41.8 | 65.2 | 70.8 | 66.5 | 37.8 | 76.0 | 42.5 | 63.9 | 58.0 | 37.5 | 42.8 | 43.8 | 31.8 | 39.0 |
| | GPTQ | 6.2 | 46.2 | 73.2 | 76.0 | 73.4 | 43.2 | 78.2 | 44.4 | 69.6 | 63.0 | 35.9 | 42.1 | 39.0 | 30.8 | 36.9 |
| | SmoothQuant+ | 5.6 | 45.0 | 71.5 | 76.8 | 73.4 | 44.6 | 76.7 | 31.9 | 67.6 | 60.9 | 32.8 | 42.7 | 40.6 | 32.4 | 37.1 |
| | QUIK | 5.0 | 47.5 | 76.4 | 78.9 | 78.4 | 42.8 | 80.4 | 46.8 | 72.5 | 65.5 | 46.1 | 57.0 | 58.2 | 40.0 | 50.3 |
| | QuaRot | 5.0 | 48.6 | 77.0 | 78.9 | 78.2 | 44.2 | 80.3 | 46.3 | 72.2 | 65.7 | 46.5 | 56.8 | 58.0 | 40.1 | 50.4 |
| | SpinQuant | 5.0 | 48.3 | 76.4 | 80.4 | 78.1 | 43.8 | 79.8 | 46.7 | 71.1 | 65.6 | 46.7 | 57.1 | 58.3 | 40.1 | 50.5 |
| | ResQ | 5.0 | 49.0 | 77.1 | 80.6 | 78.9 | 45.4 | 79.9 | 47.2 | 72.3 | 66.3 | 47.6 | 58.2 | 59.9 | 41.7 | 51.9 |
| | | | | | | | Llama 3 family | | | | | | | | | |
| Meta-Llama-3-8B | 16-bit | 6.1 | 53.2 | 77.1 | 81.1 | 79.2 | 44.8 | 80.9 | 47.0 | 73.4 | 67.1 | 55.0 | 70.6 | 73.2 | 53.7 | 63.1 |
| | RTN | 8.5 | 47.8 | 72.3 | 72.1 | 75.3 | 44.3 | 78.2 | 44.8 | 71.5 | 63.1 | 46.3 | 59.5 | 61.9 | 45.7 | 53.4 |
| | GPTQ | 7.5 | 47.1 | 71.1 | 72.2 | 72.7 | 42.6 | 78.2 | 45.7 | 72.8 | 62.8 | 40.4 | 60.3 | 61.6 | 46.2 | 52.1 |
| | SmoothQuant+ | 8.3 | 44.8 | 71.2 | 75.4 | 73.6 | 40.0 | 79.0 | 43.6 | 68.1 | 62.0 | 42.1 | 53.6 | 54.1 | 38.5 | 47.1 |
| | QUIK | 6.7 | 50.0 | 75.7 | 80.1 | 77.4 | 45.8 | 80.0 | 45.1 | 74.8 | 66.1 | 52.1 | 65.7 | 68.1 | 49.8 | 58.9 |
| | QuaRot | 6.7 | 51.6 | 78.5 | 80.0 | 77.7 | 45.2 | 79.8 | 46.4 | 73.1 | 66.5 | 51.6 | 66.8 | 68.5 | 48.7 | 58.9 |
| | SpinQuant | 6.6 | 50.0 | 77.2 | 80.3 | 77.9 | 44.0 | 80.7 | 46.6 | 72.8 | 66.2 | 52.5 | 67.2 | 68.1 | 49.5 | 59.3 |
| | ResQ | 6.5 | 54.3 | 78.6 | 77.2 | 78.4 | 44.0 | 79.2 | 46.3 | 73.2 | 66.4 | 53.6 | 68.6 | 70.0 | 52.0 | 61.0 |
| Meta-Llama-3-70B | 16-bit | 2.9 | 64.2 | 85.9 | 85.3 | 84.9 | 48.6 | 84.4 | 50.8 | 80.6 | 73.1 | 67.6 | 81.5 | 86.8 | 68.4 | 76.1 |
| | RTN | 16499 | 26.5 | 25.7 | 37.8 | 26.4 | 29.0 | 51.1 | 34.6 | 53.0 | 35.5 | 25.4 | 25.9 | 22.5 | 22.7 | 24.1 |
| | GPTQ | 8586.4 | 26.5 | 24.9 | 38.1 | 26.4 | 29.4 | 51.9 | 34.9 | 49.4 | 35.2 | 25.7 | 23.6 | 22.5 | 23.4 | 23.8 |
| | SmoothQuant+ | - | - | - | - | - | - | - | - | - | - | - | - | - | - | - |
| | QUIK | 3.7 | 60.3 | 82.0 | 83.5 | 83.5 | 45.4 | 82.4 | 47.8 | 78.1 | 70.4 | 65.2 | 79.0 | 84.1 | 65.2 | 73.4 |
| | QuaRot | 3.6 | 60.0 | 84.3 | 84.9 | 83.9 | 49.2 | 83.9 | 49.4 | 78.8 | 71.8 | 64.3 | 80.0 | 85.8 | 66.7 | 74.2 |
| | SpinQuant | - | - | - | - | - | - | - | - | - | - | - | - | - | - | - |
| | ResQ | 3.3 | 63.0 | 84.7 | 84.4 | 84.4 | 48.2 | 84.2 | 50.1 | 80.8 | 72.5 | 68.2 | 86.0 | 80.8 | 66.8 | 75.4 |
| | | | | | | | Llama 3.2 family | | | | | | | | | |
| Llama-3.2-1B | 16-bit | 9.8 | 36.5 | 60.6 | 63.4 | 63.6 | 37.4 | 74.5 | 42.8 | 60.1 | 54.9 | 34.8 | 41.1 | 39.9 | 32.0 | 36.9 |
| | RTN | 16.6 | 30.6 | 46.7 | 61.9 | 55.2 | 32.4 | 66.7 | 38.0 | 56.7 | 48.5 | 25.1 | 26.7 | 26.1 | 25.2 | 25.8 |
| | GPTQ | 15.3 | 32.8 | 50.9 | 61.6 | 54.8 | 31.6 | 67.4 | 39.1 | 55.8 | 49.2 | 24.1 | 26.2 | 23.9 | 24.3 | 24.6 |
| | SmoothQuant+ | 20.6 | 30.0 | 47.7 | 50.2 | 50.8 | 31.2 | 66.3 | 37.5 | 54.1 | 46.0 | 24.9 | 26.6 | 25.5 | 23.9 | 25.2 |
| | QUIK | 11.6 | 35.0 | 57.9 | 62.3 | 59.4 | 35.4 | 71.7 | 41.9 | 56.9 | 52.6 | 28.2 | 31.5 | 29.6 | 27.2 | 29.1 |
| | QuaRot | 11.1 | 34.1 | 58.8 | 52.3 | 59.8 | 36.4 | 72.3 | 41.2 | 58.6 | 51.7 | 28.3 | 30.6 | 29.5 | 26.4 | 28.7 |
| | SpinQuant | 11.1 | 34.2 | 55.6 | 61.8 | 60.0 | 35.0 | 72.1 | 40.8 | 57.6 | 52.1 | 26.1 | 27.7 | 27.4 | 24.0 | 26.3 |
| | ResQ | 10.4 | 34.8 | 58.0 | 62.2 | 61.1 | 34.4 | 72.3 | 41.8 | 60.1 | 53.1 | 31.3 | 36.0 | 36.2 | 31.0 | 33.6 |
| Llama-3.2-3B | 16-bit | 7.8 | 46.2 | 71.7 | 73.1 | 73.7 | 43.4 | 77.4 | 47.2 | 69.1 | 62.7 | 48.9 | 62.9 | 62.3 | 45.2 | 54.8 |
| | RTN | 17.8 | 36.6 | 51.3 | 55.3 | 64.3 | 36.6 | 73.5 | 42.1 | 62.4 | 52.8 | 38.5 | 46.7 | 46.6 | 35.0 | 41.7 |
| | GPTQ | 14.1 | 37.0 | 59.9 | 57.3 | 62.9 | 36.8 | 74.3 | 41.9 | 64.4 | 54.3 | 37.1 | 47.7 | 46.2 | 36.5 | 41.9 |
| | SmoothQuant+ | 12.7 | 37.0 | 65.9 | 53.3 | 61.9 | 34.8 | 71.2 | 41.6 | 63.2 | 52.2 | 31.4 | 37.6 | 40.7 | 32.4 | 35.5 |
| | QUIK | 8.6 | 42.1 | 65.9 | 71.8 | 71.7 | 40.0 | 76.0 | 44.6 | 66.7 | 59.8 | 45.2 | 57.2 | 57.8 | 40.7 | 50.2 |
| | QuaRot | 8.4 | 43.4 | 68.9 | 69.5 | 71.2 | 40.6 | 76.8 | 46.0 | 67.2 | 60.5 | 45.0 | 56.1 | 56.0 | 40.0 | 49.3 |
| | SpinQuant | 8.4 | 43.5 | 67.8 | 70.6 | 71.9 | 41.6 | 76.9 | 44.9 | 68.5 | 60.7 | 46.1 | 56.7 | 57.1 | 39.4 | 49.8 |
| | ResQ | 8.1 | 44.4 | 69.4 | 72.4 | 72.2 | 41.8 | 76.3 | 45.2 | 69.1 | 61.3 | 48.2 | 61.1 | 59.8 | 44.5 | 53.4 |

of sequence length 2048 taken from Wikitext. For complete training hyperparameters, we guide the interested readers to official implementation of SpinQuant (Liu et al., 2025).

## I. Complete results with different calibration datasets

In Table 16, we provide results on wikitext perplexity, task-by-task accuracy on 0-shot reasoning benchmarks and task-by-task accuracy on MMLU benchmarks when obtaining ResQ projection matrix $P$ using different calibration datasets. We find no clear consensus on the optimality of one particular dataset. The performance results for different datasets show no

Table 13: Accuracy (higher is better) on 0-shot massive multi-discipline multimodal understanding and reasoning tasks across six subjects: Art & Design, Business, Science, Health & Medicine, Humanities & Social Science, and Tech & Engineering for the Qwen2 VL Instruct family when quantized to **W/A/KV = 4/4/4** bits and **W/A/KV = 4/8/4** bits. Results of all techniques were obtained using their official codebase. Our work ResQ and QUIK (Ashkboos et al., 2024b) keep $1/8$ of channels in 8-bit. All techniques except RTN use GPTQ (Frantar et al., 2023).

| | | Qwen2-VL-2B-Instruct | | | | | | |
|---|---|---|---|---|---|---|---|---|
| **W/A/KV** (bit) | Method | 0-shot MMMU tasks | | | | | | |
| | | Art-Design | Business | Science | Health | Humanities | Tech | Avg. |
| 16/16/16 | Baseline | 56.7 | 36.0 | 37.3 | 50.8 | 26.0 | 31.0 | 39.6 |
| | RTN | 28.3 | 18.7 | 26.0 | 26.7 | 21.3 | **29.1** | 25.0 |
| 4/4/4 | GPTQ | 28.3 | **27.3** | 27.0 | 29.0 | **26.7** | 27.6 | 27.7 |
| | QuaRot | 24.2 | 23.3 | 20.7 | 26.7 | 26.0 | 22.9 | 24.0 |
| | QUIK | 25.8 | 26.0 | 26.7 | 29.2 | 26.0 | 24.3 | 26.3 |
| 4.5/4.5/4.5 | ResQ | **38.3** | 21.3 | **28.7** | **45.0** | 21.3 | 23.3 | **29.7** |
| | RTN | 27.5 | 21.3 | 27.3 | 24.2 | 21.3 | 27.6 | 24.9 |
| 4/8/4 | GPTQ | 24.2 | 23.3 | 24.0 | 18.3 | 21.3 | 29.5 | 23.4 |
| | QuaRot | 20.0 | 24.7 | 30.0 | 26.7 | 26.0 | **31.4** | 26.5 |
| | QUIK | 33.3 | 28.7 | 32.0 | 32.5 | 26.0 | 18.1 | 28.4 |
| 4.5/8/4.5 | ResQ | **37.5** | **32.0** | **32.7** | **47.5** | **26.7** | 27.6 | **34.0** |

| | | Qwen2-VL-7B-Instruct | | | | | | |
|---|---|---|---|---|---|---|---|---|
| **W/A/KV** (bit) | Method | 0-shot MMMU tasks | | | | | | |
| | | Art-Design | Business | Science | Health | Humanities | Tech | Avg. |
| 16/16/16 | Baseline | 68.3 | 41.3 | 54.7 | 68.3 | 38.7 | 38.1 | 51.6 |
| | RTN | 24.2 | 28.0 | 29.3 | 22.5 | 29.3 | 27.1 | 26.7 |
| 4/4/4 | GPTQ | 21.7 | 26.0 | 25.3 | 28.3 | 24.7 | 23.3 | 24.9 |
| | QuaRot | 21.7 | 21.3 | 28.7 | 25.0 | 20.7 | 29.5 | 24.5 |
| | QUIK | 30.8 | 30.0 | 32.0 | 26.7 | 28.0 | 26.2 | 28.9 |
| 4.5/4.5/4.5 | ResQ | **65.0** | **39.3** | **45.3** | **61.7** | **34.0** | **36.7** | **47.0** |
| | RTN | 23.3 | 28.7 | 27.3 | 25.0 | 22.7 | 24.3 | 25.2 |
| 4/8/4 | GPTQ | 20.8 | 23.3 | 30.0 | 19.2 | 24.0 | 28.6 | 24.3 |
| | QuaRot | 20.8 | 26.0 | 30.0 | 19.2 | 24.7 | 26.2 | 24.5 |
| | QUIK | 25.0 | 23.3 | 31.3 | 26.7 | 25.3 | 26.7 | 26.4 |
| 4.5/8/4.5 | ResQ | **67.5** | **39.3** | **51.3** | **64.2** | **36.7** | 33.8 | **48.8** |

significant fluctuations.

## J. Artifact licenses

According to their licenses, all language models used in the paper fall under acceptable use case. The licenses for the models are linked for perusal: `Llama-2-7b-hf`, `Llama-2-13b-hf`, `Meta-Llama-3-8B`, `Meta-Llama-3-70B`, `Llama-3.2-1B`, `Llama-3.2-3B`, `Qwen2.5-0.5B`, `Qwen2.5-1.5B`, `Qwen2.5-3B`, `Qwen2.5-7B`, `Qwen2.5-14B`, `Qwen2.5-32B`, `Qwen2.5-72B`, `Qwen2-VL-7B-Instruct`, and `Qwen2-VL-2B-Instruct`.

Table 14: Comparison of ResQ, QuaRot, SpinQuant at **iso-bitwidth of W/A/KV = 4/4/4**. ResQ keeps ⅛ channels with high eigen value in 6-bit, ⅛ channels with low eigen value in 2-bit and remaining in 4-bit for average bit-width of 4-bit. Evaluations include perplexity on Wikitext, accuracy on eight 0-shot common sense reasoning tasks including ARC-challenge, ARC-easy, BoolQ, HellaSwag, Openbook QA, PIQA, SIQA, and WinoGrande, and on 0-shot massive multitask language understanding tasks across four subjects: STEM, Humanities, Social Sciences, and MMLU-other, for the Qwen2.5 model family. (↓): lower is better, (↑): higher is better.

| Model | Method | Perplexity Wiki (↓) | ARC-c (↑) | ARC-e (↑) | BoolQ (↑) | HellaS (↑) | OBQA (↑) | PIQA (↑) | SIQA (↑) | WinoG (↑) | Avg. (↑) | humanities (↑) | Other (↑) | SocialS (↑) | STEM (↑) | Avg. (↑) |
|---|---|---|---|---|---|---|---|---|---|---|---|---|---|---|---|---|
| Qwen2.5-3B | QuaRot | 68.8 | 32.4 | 53.1 | 51.6 | 49.2 | 33.4 | 66.7 | 39.3 | 56.4 | 47.7 | 28.1 | 32.0 | 28.9 | 26.6 | 28.9 |
|  | SpinQuant | 70.6 | 33.3 | 56.4 | 51.8 | 49.5 | 33.0 | 66.7 | 41.0 | 57.1 | 48.6 | 30.1 | 37.2 | 33.9 | 30.1 | 32.8 |
|  | ResQ | **9.8** | **44.1** | **70.2** | **70.2** | **67.3** | **41.0** | **73.7** | **45.4** | **61.1** | **59.1** | **45.8** | **58.4** | **58.9** | **45.9** | **52.2** |
| Qwen2.5-7B | QuaRot | 4035.9 | 25.9 | 41.0 | 39.1 | 29.1 | 27.6 | 57.9 | 35.7 | 50.6 | 38.4 | 24.8 | 24.4 | 24.4 | 22.7 | 24.1 |
|  | SpinQuant | 3395.4 | 25.5 | 44.2 | 39.5 | 29.3 | 27.0 | 58.8 | 35.3 | 49.4 | 38.6 | 25.0 | 23.8 | 24.2 | 24.3 | 24.3 |
|  | ResQ | **34.2** | **42.6** | **60.2** | **75.4** | **59.4** | **41.6** | **63.1** | **44.6** | **62.8** | **56.2** | **49.3** | **63.6** | **68.6** | **50.4** | **58.0** |
| Qwen2.5-14B | QuaRot | 6.8 | 54.8 | 79.6 | 79.9 | 78.7 | 44.0 | 79.5 | 49.9 | **70.7** | 67.1 | 60.9 | 75.1 | **80.2** | 67.3 | 70.9 |
|  | SpinQuant | 6.6 | 54.2 | 81.2 | **82.1** | 77.9 | **44.6** | 78.4 | 50.8 | 70.0 | 67.4 | 59.6 | **75.3** | 78.9 | 66.7 | 70.1 |
|  | ResQ | **6.5** | **55.6** | **81.3** | 79.2 | **79.2** | 43.6 | **79.9** | **51.4** | 70.2 | **67.5** | **61.6** | **75.3** | 80.1 | **68.2** | **71.3** |
| Qwen2.5-32B | QuaRot | 6.1 | 54.3 | 78.6 | 83.0 | 81.0 | 43.4 | 79.8 | 50.6 | 71.7 | 67.8 | 67.4 | 80.5 | 85.2 | 75.1 | 77.0 |
|  | SpinQuant | 6.0 | 54.5 | 76.1 | 85.1 | 81.5 | 44.2 | 80.1 | 51.3 | 70.4 | 67.9 | **68.5** | 80.0 | 86.0 | 76.0 | 77.6 |
|  | ResQ | **5.9** | **55.9** | **79.7** | **85.8** | **81.6** | **44.8** | **80.3** | **51.6** | **73.4** | **69.1** | 68.4 | **80.6** | **86.4** | **76.2** | **77.9** |
| Qwen2.5-72B | QuaRot | 4.9 | 55.8 | 81.1 | **87.5** | 84.0 | 45.2 | 81.7 | 52.5 | 74.5 | 70.3 | **71.4** | 84.2 | **87.7** | 77.1 | 80.1 |
|  | ResQ | 4.9 | **58.8** | **81.9** | 87.2 | **84.1** | **46.0** | **82.2** | **53.2** | **75.5** | **71.1** | 71.3 | 84.2 | 87.4 | **77.4** | 80.1 |

Table 15: Comparison of ResQ and ResQ+training rotation R, evaluating perplexity on Wikitext, accuracy on eight 0-shot common sense reasoning tasks including ARC-challenge, ARC-easy, BoolQ, HellaSwag, Openbook QA, PIQA, SIQA, and WinoGrande, and on 0-shot massive multitask language understanding tasks across four subjects: STEM, Humanities, Social Sciences, and MMLU-other, for the Meta-Llama-3-8B, Llama-2-7b-hf, Qwen2.5-7B, and Llama-3.2-1B models. (↓): lower is better, (↑): higher is better.

| Model | Method | Perplexity Wiki (↓) | ARC-c (↑) | ARC-e (↑) | BoolQ (↑) | HellaS (↑) | OBQA (↑) | PIQA (↑) | SIQA (↑) | WinoG (↑) | Avg. (↑) | humanities (↑) | Other (↑) | SocialS (↑) | STEM (↑) | Avg. (↑) |
|---|---|---|---|---|---|---|---|---|---|---|---|---|---|---|---|---|
| Meta-Llama-3-8B | ResQ | 7.1 | **49.2** | 75.0 | 72.5 | **76.5** | 43.0 | **78.3** | 45.8 | **71.0** | 63.9 | 50.6 | 64.4 | 65.8 | 48.1 | 57.2 |
|  | ResQ + training R | **7.0** | 48.1 | **76.1** | **76.7** | 76.4 | **44.2** | 77.8 | 45.7 | 70.6 | **64.5** | **51.0** | **65.9** | **67.4** | **48.9** | **58.3** |
| Llama-2-7b-hf | ResQ | 5.8 | 44.0 | **72.6** | **75.3** | 74.0 | 41.0 | **77.9** | 43.9 | 66.9 | 62.0 | **35.9** | 40.9 | **42.1** | 32.1 | 37.7 |
|  | ResQ + training R | 5.8 | **44.9** | 71.6 | 74.9 | **74.5** | **42.0** | 77.6 | **44.6** | **67.4** | **62.2** | 35.4 | **42.7** | 40.4 | **33.4** | **38.0** |
| Qwen2.5-7B | ResQ | 8.2 | 49.0 | 74.7 | **81.4** | 75.7 | **45.0** | **78.9** | 49.4 | 68.2 | 65.3 | 57.8 | **74.4** | **79.3** | 64.5 | **69.0** |
|  | ResQ + training R | **8.0** | **50.0** | **75.2** | 80.9 | **76.1** | 44.0 | 78.7 | **52.3** | **69.2** | **65.8** | **59.0** | 74.0 | 79.0 | **65.0** | **69.0** |
| Llama-3.2-1B | ResQ | 12.4 | **34.0** | **54.2** | 57.0 | 57.3 | 31.2 | 69.4 | **41.0** | **56.8** | **50.1** | 28.3 | 30.5 | **31.3** | 27.6 | 29.4 |
|  | ResQ + training R | **11.7** | 32.9 | 53.9 | **57.5** | **58.2** | **33.6** | **70.1** | 39.4 | 55.6 | **50.1** | **29.0** | **30.8** | 30.6 | **28.0** | **29.6** |

Table 16: Performance of ResQ with different calibration datasets, evaluating perplexity on Wikitext, accuracy on eight 0-shot common sense reasoning tasks including ARC-challenge, ARC-easy, BoolQ, HellaSwag, Openbook QA, PIQA, SIQA, and WinoGrande, and on 0-shot massive multitask language understanding tasks across four subjects: STEM, Humanities, Social Sciences, and MMLU-other, for the Llama-3.2-3B, Meta-Llama-3-8B, Qwen2.5-3B, and Qwen2.5-7B models. (↓): lower is better, (↑): higher is better.

| Model | Method | Perplexity Wiki (↓) | ARC-c (↑) | ARC-e (↑) | BoolQ (↑) | HellaS (↑) | OBQA (↑) | PIQA (↑) | SIQA (↑) | WinoG (↑) | Avg. (↑) | humanities (↑) | Other (↑) | SocialS (↑) | STEM (↑) | Avg. (↑) |
|---|---|---|---|---|---|---|---|---|---|---|---|---|---|---|---|---|
| Llama-3.2-3B | Wikitext | **8.8** | **43.1** | 65.6 | 68.8 | 70.5 | 38.4 | 75.1 | 45.6 | 64.8 | 59.0 | **44.7** | **57.0** | **56.5** | **41.0** | **49.8** |
|  | C4 | **8.8** | 42.1 | **66.7** | 68.9 | **70.6** | **41.2** | 75.4 | 45.3 | 64.2 | **61.7** | 43.9 | 55.5 | 55.0 | 40.2 | 48.6 |
|  | PTB | **8.8** | 41.8 | 65.8 | **70.2** | 70.0 | 39.2 | 74.9 | **46.1** | 65.2 | 59.1 | 43.5 | 54.8 | 53.1 | 38.9 | 47.6 |
|  | Alpaca | **8.8** | **43.1** | 65.9 | 66.6 | 70.4 | 38.8 | **75.5** | 44.9 | **66.2** | 58.9 | 43.6 | 54.8 | 53.9 | 39.6 | 48.0 |
| Meta-Llama-3-8B | Wikitext | **7.1** | **49.2** | 75.0 | 72.5 | 76.5 | **43.0** | 78.3 | 45.8 | **71.0** | 63.9 | 50.6 | 64.4 | 65.8 | **48.1** | 57.2 |
|  | C4 | **7.1** | 48.0 | **76.3** | 76.6 | 76.4 | 42.4 | **78.5** | **46.0** | 68.1 | **64.0** | **50.8** | 64.6 | 66.7 | 47.1 | 57.3 |
|  | PTB | **7.1** | 48.1 | 73.3 | **78.1** | 76.4 | 42.6 | 77.6 | 45.2 | 70.3 | 63.9 | 49.6 | 63.5 | 65.0 | 47.0 | 56.3 |
|  | Alpaca | **7.1** | 47.8 | 73.1 | 76.8 | **76.9** | 42.8 | 77.8 | 45.1 | **71.0** | 63.9 | 50.4 | **65.5** | **67.0** | 47.3 | **57.5** |
| Qwen2.5-3B | Wikitext | **9.0** | 45.3 | 70.5 | 72.7 | 70.2 | **42.4** | 76.8 | 46.7 | 64.4 | 61.1 | **53.1** | **66.5** | 70.5 | 54.8 | **61.2** |
|  | C4 | **9.0** | 44.2 | 70.4 | 71.6 | **70.6** | 40.6 | 76.3 | **46.9** | 65.0 | 60.7 | 51.8 | 65.5 | 68.2 | 52.7 | 59.6 |
|  | PTB | 9.1 | 42.4 | 68.5 | 70.5 | 69.7 | 40.6 | 75.5 | 46.3 | 64.5 | 59.7 | 51.2 | 65.6 | 68.4 | 52.8 | 59.5 |
|  | Alpaca | **9.0** | **46.2** | **72.5** | **74.5** | 70.4 | 38.6 | 76.4 | 46.6 | **65.4** | **61.3** | 52.2 | 65.1 | **71.1** | **55.3** | 60.9 |
| Qwen2.5-7B | Wikitext | 8.2 | 49.0 | 74.7 | 81.4 | 75.7 | 45.0 | **78.9** | 49.4 | **68.2** | 65.3 | 57.8 | **74.4** | **79.3** | 64.5 | **69.0** |
|  | C4 | 8.2 | **50.4** | **75.9** | **82.2** | 75.9 | 42.8 | 78.6 | 52.3 | 67.5 | **65.7** | 58.4 | 72.7 | 79.0 | 64.1 | 68.5 |
|  | PTB | **8.0** | 47.8 | 74.5 | 81.8 | **76.3** | **45.6** | 77.2 | 52.7 | 66.5 | 65.3 | 58.1 | 74.0 | 78.7 | 64.4 | 68.8 |
|  | Alpaca | 8.9 | 50.3 | 75.0 | 82.0 | 75.8 | 43.2 | 78.2 | **52.9** | 68.1 | **65.7** | **58.8** | 73.3 | 78.8 | 62.9 | 68.4 |

