# OpenReview forum: "ResQ: Mixed-Precision Quantization of Large Language Models with Low-Rank Residuals"
_ICML.cc/2025/Conference — ICML 2025 spotlightposter_

### Official Review · Reviewer_Azbj · 2025-03-12

**Overall Recommendation:** 1

**Summary:**

This paper introduces ResQ, a post-training quantization (PTQ) method for large language models (LLMs) that enables mixed-precision quantization of weights, activations, and KV caches. Experimental results demonstrate that ResQ achieves superior performance compared to existing methods.

**Claims And Evidence:**

The paper claims the development of custom CUDA kernels as a main contribution. However, this claim is not adequately supported. There is no detailed description of these kernels in the manuscript, and the anonymous code link does not appear to contain their implementation or sufficient details to verify this claim.

Furthermore, the fast hadamard transformation is well-established in prior work (e.g., https://github.com/Dao-AILab/fast-hadamard-transform). The paper does not clearly propose any novel contribution.

**Essential References Not Discussed:**

The paper appears to comprehensively cover the relevant related works and does not seem to have omitted any essential references.

**Experimental Designs Or Analyses:**

The experimental design generally follows established practices in the field.

**Methods And Evaluation Criteria:**

The choice of mixed-precision quantization and a rotation-based approach is reasonable, given the established effectiveness of both techniques in addressing quantization challenges. However, the novelty of combining these specific approaches is not fully confirmed.

**Other Comments Or Suggestions:**

A significant claim made in the paper is the development of custom CUDA kernels for performance acceleration. However, this claim is not supported by sufficient evidence within the paper. There is no dedicated section or subsection detailing the design, implementation, or optimization strategies employed in these kernels.

**Other Strengths And Weaknesses:**

A primary concern is the limited novelty of the proposed approach. While the paper combines mixed-precision quantization and a rotation-based method, this combination lacks significant new insights. The work feels more like a technical report demonstrating the application of existing techniques rather than a substantial research contribution.

**Questions For Authors:**

Figure 4 in the submission appears to be highly similar (**only color changed?**) in structure and content to Figure 1 in SpinQuant paper (https://arxiv.org/pdf/2405.16406). Could the authors clarify the relationship between these two figures and explain any differences or novel aspects of Figure 4 in the current work?

**Relation To Broader Scientific Literature:**

The key contributions of the paper do not appear to introduce significant new findings or ideas compared to the existing scientific literature.

**Theoretical Claims:**

The proofs for the theoretical claims in this paper are correct.

---

> ### Author Rebuttal · Authors · 2025-03-31
>
> Thank you, Reviewer Azbj, for putting effort in reviewing our paper. We provide a response to your concerns below.
>
> 1. **Limited Novelty:** While we agree with the reviewer regarding the statement that ResQ is a rotation and mixed precision quantization based approach, we respectfully disagree regarding the novelty aspect of our work. To the best of our knowledge, there is no work which does what ResQ does and we would be interested if the reviewer can point us to the related paper. In Table 4 of main paper, we also show that ResQ outperforms a hypothetical outlier+rotation based approach which is an amalgamation of related baselines QUIK (EMNLP2024) and QuaRot (NeurIPS2024). This highlights optimality of PCA based high precision component extraction which again to the best of knowledge has not been explored before in literature. Additionally, we provide even more comprehensive comparison in the Table 10 below where we compare ResQ and outlier+rotation baseline's Reasoning and MMLU accuracy in addition to Wikitext perplexity for Qwen2.5 models. Emphasizing that **ResQ outperforms the outlier+rotation baseline across the stack**. Based on these insights we humbly request the reviewer to re-evaluate their stance on novelty of our work.
>
> Table 10: Comparison of ResQ with outlier+rotation approach.
> | Model | Method | Wiki. PPL | Avg. Reasoning acc. | Avg. MMLU acc. |
> |:---:|:---:|:---:|:---:|:---:|
> | Qwen2.5-3B | outlier+rot. | 9.4 | 58.4 | 59.2 |
> |  | ResQ | **9.0** | **61.1** | **61.2** |
> | Qwen2.5-7B | outlier+rot. | 10.5 | 64.1 | 64.9 |
> |  | ResQ | **8.2** | **65.3** | **69.0** |
> | Qwen2.5-14B | outlier+rot. | 6.4 | 68.0 | 73.8 |
> |  | ResQ | **6.2** | **69.2** | **74.6** |
>
> Detailed task-wise results for the above table can be found [here](https://shorturl.at/KTiZR) in Table 4.
>
> 2. **CUDA Kernel:** Thank you for highlighting this point. We will include additional details about the CUDA kernel in the main paper and plan to release the code upon acceptance. The CUDA Kernel involves, the mixed precision quantization of activations into 4/8-bit components within a single kernel, low precision GEMM of 4-bit and 8-bit operands and a fused dequantization of 4-bit and 8-bit results. Further, the hardware implementation involves quantizing and packing the mixed precision kv cache for memory efficiency. Additionally, we provide several new hardware-related results to address the reviewer's concerns: improved memory usage with ResQ (Table 5, response to Reviewer vpAp), speedup achieved by ResQ on long context lengths (Table 2, response to Reviewer UnBL), inference latency in a distributed setting representative of datacenter workloads (Table 7, response to Reviewer vpAp), and a comparison of ResQ with a baseline quantization kernel (Table 5, response to Reviewer vpAp). We hope these additions comprehensively address the concerns regarding the CUDA kernel.
>
> 3. **Figure 4:** ResQ and SpinQuant project weights and activations by projection matrices in similar manner. While SpinQuant trains the projection matrices and utilizes uniform precision quantization across the layers, ResQ uses PCA basis and random orthogonal rotations as projection matrices and performs mixed precision quantization. The key differentiation in Figure 4 is the layers and activations which are quantized to mixed precision. For ResQ all the layers and activations except down\_proj layer is mixed precision.

---

### Official Review · Reviewer_x9DL · 2025-03-14

**Overall Recommendation:** 4

**Summary:**

The paper presents ResQ, a mixed-precision post-training quantization (PTQ) method for large language models (LLMs). The core idea of ResQ is to compute the orthogonal transformations using PCA and decompose the orthogonal matrices for high-precision and low-precision based on their corresponding eigenvalues. Moreover, high-precision weights and activations are cast in 8-bit. Experiments demonstrate that ResQ outperforms strong baselines such as SpinQuant, QuaRot, and QUIK.

**Update after rebuttal**: My latest reply reflected my final update.

**Claims And Evidence:**

[Correct Claims]

* Custom cuda kernels to speed up inference.
* Theoretical analysis on PCA-based projections.
* ResQ outperforms recent uniform and mixed precision PTQ methods. -> partially correct and I have some comments in Weakness in Experimental Designs Or Analyses.

[Problematic Claims]

* In line 29, the paper claims using PCA to compute orthogonal matrices is "provably optimal mixed precision quantization scheme". However, in theorem 4.2, the method is optimal in minimizing layer loss, but it may not be optimal on the final output loss, and the latter is done in SpinQuant. Therefore, it would be great to tune down this claim a bit, and I am curious why ResQ performs better than SpinQuant (please see my comment in Weakness in Experimental Designs Or Analyses).

**Essential References Not Discussed:**

N/A

**Experimental Designs Or Analyses:**

[Strength]

* The experiments are conducted on multiple models and benchmarks.

[Weakness]

* For the compared methods, some are uniform and some are mixed-precision quantization, and it is unclear how many additional bits the mixed-precision methods used. There should be a column in the tables about the number of bits for each method for a fair comparison. It is also fairer to use the same bitness for all the approaches, and maybe the authors can adjust the number of bits for weights or the scale and zeros that are introduced in quantization to do it.

**Methods And Evaluation Criteria:**

* The datasets and baselines make sense.

**Other Comments Or Suggestions:**

* As the PCA part is the main novelty of this paper, it would be great to have a standalone section or at least a paragraph to introduce it. I spent some time to finally find it is in Sec 4.2.
* In line 219, defining how to compute the quantization SNR would be great as it seems to pop out from nowhere here.

**Other Strengths And Weaknesses:**

N/A

**Questions For Authors:**

N/A

**Relation To Broader Scientific Literature:**

* PCA for orthogonal transformation -> related to low-rank decomposition, like LoRA and other rotation methods (QuaRot, SpinQuant).
* The method combines rotation and mixed-precision quantization -> which is an interesting combination from previous methods.

**Theoretical Claims:**

I didn't check the proof of Theorem 4.2.

---

> ### Author Rebuttal · Authors · 2025-03-31
>
> Thank you, Reviewer x9DL, for your effort in reviewing our paper. We appreciate your recognition of the strength of ResQ's experimental section. Below, we respond to the concerns you raised.
>
> 1. **Claim regarding quantization error:** ResQ's approach of choosing coefficients along principal eigen vectors of activations to keep in 8-bit is indeed **optimal to minimize local quantization error**. Projection with PCA basis $P$ minimizes quantization error across high/low precision groups while random orthogonal matrix $R$ improves quantization error within high/low precision group. We will further clarify this point in the main paper. While SpinQuant learns rotation matrices minimizing final loss, it still keeps all the components in 4-bit which does not minimize quantization error enough while ResQ intelligently performs mixed precision quantization. Moreover, SpinQuant's training of rotation matrices can additionally be incorporated in ResQ as well at the cost of high calibration time. The 8-bit components can be chosen via PCA basis ($P$) and orthogonal matrix to minimize quantization error within 4-bit and 8-bit quantization groups ($R$) can be learned minimizing output loss. Such an approach further improves performance of ResQ as shown in Table 8 below surpassing SpinQuant by even higher amount.
>
> Table 8: Performance of ResQ and variant of ResQ which trains rotation $R$.
> | Model | Method | Wiki. PPL | Avg. Reasoning acc. | Avg. MMLU acc. |
> |:---:|:---:|:---:|:---:|:---:|
> | Meta-Llama-3-8B | ResQ | 7.1 | 63.9 | 57.2 |
> |  | ResQ + training $R$ | **7.0** | **64.5** | **58.3** |
> | Llama-2-7B | ResQ | **5.8** | 62.0 | 37.7 |
> |  | ResQ + training $R$ | **5.8** | **62.2** | **38.0** |
> | Qwen2.5-7B | ResQ | 8.2 | 65.3 | **69.0** |
> |  | ResQ + training $R$ | **8.0** | **65.8** | **69.0** |
> | Llama-3.2-1B | ResQ | 12.4 | **50.1** | 29.4 |
> |  | ResQ + training $R$ | **11.7** | **50.1** | **29.6** |
>
> Detailed task-wise results for the above table can be found [here](https://shorturl.at/KTiZR) in Table 3.
>
> 2. **Iso-bitwidth comparison:** Both ResQ and previous state of the art mixed precision quantization technique, QUIK (EMNLP2024), keep $\frac{1}{8}$ channels in 8-bit which brings average bit-width to 4.5 bits. For other baselines, achieving fractional bit-width is impossible with uniform precision quantization across layers and channels. Additionally, as suggested by the reviewer, we perform comparison at equal bit-width of 4-bit. We create a variant of ResQ which keeps first $\frac{d}{8}$ channels corresponding to lowest eigen value of $P$ in 2-bit, $\frac{d}{8}$ channels corresponding to highest eigen value of $P$ in 6-bit and remaining in 4-bit to achieve average bit-width of 4-bit. As shown in Table 9 below, **even at iso-bitwidth of 4-bit, ResQ outperforms SpinQuant and QuaRot** highlighting its capabilities.
>
> Table 9: Iso-bitwidth comparison between ResQ, QuaRot and SpinQuant.
> | Model | Method | Wiki. PPL | Avg. Reasoning acc. | Avg. MMLU acc. |
> |:---:|:---:|:---:|:---:|:---:|
> | Qwen2.5-3B | QuaRot | 68.8 | 47.7 | 28.9 |
> |  | SpinQuant | 70.6 | 48.6 | 32.8 |
> |  | ResQ | **9.8** | **59.1** | **52.2** |
> | Qwen2.5-7B | QuaRot | 4e3 | 38.4 | 24.1 |
> |  | SpinQuant | 3e3 | 38.6 | 24.3 |
> |  | ResQ | **34.2** | **56.2** | **58.0** |
> | Qwen2.5-14B | QuaRot | 6.8 | 67.1 | 70.9 |
> |  | SpinQuant | 6.6 | 67.4 | 70.1 |
> |  | ResQ | **6.5** | **67.5** | **71.3** |
> | Qwen2.5-32B | QuaRot | 6.1 | 67.8 | 77.0 |
> |  | SpinQuant | 6.0 | 67.9 | 77.6 |
> |  | ResQ | **5.9** | **69.1** | **77.9** |
> | Qwen2.5-72B | QuaRot | **4.9** | 70.3 | 80.1 |
> |  | ResQ | **4.9** | **71.1** | **80.1** |
>
> Detailed task-wise results for the above table can be found [here](https://shorturl.at/KTiZR) in Table 5.

---

> > ### Comment · Reviewer_x9DL · 2025-04-05
> >
> > Thanks the authors for the rebuttal contents, which addressed my concerns about fairer bit comparison. I will increase the rating accordingly.

---

### Official Review · Reviewer_vpAp · 2025-03-15

**Overall Recommendation:** 4

**Summary:**

The paper proposes a novel algorithm to separate high/low values and respectively smooth and quantize them with different precisions. Theoretical analyses suggest that by introducing a designed matrix $P$, the upper bound of the error can be minimized. The experimental results illustrate the effectiveness and efficiency of the proposed method.

**Claims And Evidence:**

The claims made are supported by clear and convincing evidence.

**Essential References Not Discussed:**

To the best of my knowledge, there are no missing references.

**Experimental Designs Or Analyses:**

The experiment is well-structured. However, from my perspective, there are two issues that should be addressed:

1. Memory reduction compared to FP16, INT4, and/or other baseline methods should be provided.

2. Is it possible to validate speedup on other NVIDIA GPUs, such as the A100?

**Methods And Evaluation Criteria:**

The methods and evaluation criteria are appropriate for the problem.

**Other Comments Or Suggestions:**

No additional comments or suggestions.

**Other Strengths And Weaknesses:**

Strengths:
1. The proposed method is novel and efficient. The use of the PCA method addresses the difficulties in separating high/low values in $X$, contributing to the success of the mixed-precision quantization method, which is quite impressive.

2. The experiment is solid and persuasive in supporting the proposed method. The results are strong, and the overhead is acceptable.

3. The paper is well-structured, making it easy for readers to follow.

Btw, this is quite an impressive paper I have reviewed.

Weakness:

Aside from the two issues mentioned in the "Experimental Designs or Analyses" section, there is one more weakness: The authors should try to provide more details regarding the ResQ kernel. Is it a simple combination of INT4 and INT8 kernels with an addition? If so, I believe that developing CUDA kernels alone can hardly be considered a main contribution in the introduction section.

**Questions For Authors:**

See weaknesses above.

**Relation To Broader Scientific Literature:**

The key contributions can be summarized as follows: The proposed method employs PCA-like permutation and rotation matrices to effectively mitigate outliers in WA quantization and enhance performance, surpassing SOTA methods such as QuaRot and SpinQuant. The proposed method is also validated through speedup comparisons, further illustrating its efficiency, leading to the development of (mixed-precision) WAKV quantization.

**Theoretical Claims:**

I have not carefully checked the proofs of the theoretical claims. However, the claims make sense and are likely to hold.

---

> ### Author Rebuttal · Authors · 2025-03-31
>
> Thank you, Reviewer vpAp, for your effort in reviewing our paper. We are delighted that you find our approach impressive. We provide response to your questions below.
>
> 1. **Details about compute kernel**: The compute kernel goes beyond simple combination of INT4 and INT8 kernels. More precisely, the mixed precision quantization of activations into 4/8-bit components is handled by a single kernel call as opposed to two calls to the quantization kernel. Further, the hardware implementation involves quantizing and packing the mixed precision kv cache for memory efficiency. We will provide more details in the camera ready version of the paper and will release the code upon acceptance of the paper. Additionally, we compare the compute kernel with simple combination of INT4 and INT8 kernels as mentioned by the reviewer below (Table 5 below). **The inference latency with ResQ is upto 1.3x lesser than simple combination of INT4 and INT8 kernels.**
>
> Table 5: Per decoder latency (in ms) of ResQ over simple compute kernel.
> | Model | Seq\_len | Simple | ResQ | Improv. |
> |:---:|:---:|:---:|:---:|:---:|
> | Llama-3.2-3B | 512 | 1.8 | 1.3 | 1.33 |
> |  | 8192 | 27.6 | 20.5 | 1.34 |
> | Meta-Llama-3-8B | 512 | 2.6 | 2.0 | 1.3 |
> |  | 8192 | 40.9 | 31.9 | 1.29 |
> | Qwen2.5-72B | 512 | 5.9 | 5.0 | 1.16 |
> |  | 8192 | 95.3 | 85.5 | 1.11 |
>
> 2. **Memory reduction with ResQ:** We show memory usage of ResQ on RTX 3090 (24GB), FP16 baseline and Quarot (INT4) baseline at different sequence length in Table 6 below. **ResQ consumes 1.84-3.08x lower memory than FP16 baseline** and requires slightly (4-11\%) more memory compared to QuaRot. Further, ResQ supports inference of Qwen2.5-14B where FP16 baseline runs into OOM error.
>
> Table 6: Memory of ResQ and baselines on RTX 3090.
> | Model | seq\_len | FP16 | QuaRot | ResQ | Improv.  over FP16 | Improv. over QuaRot |
> |:---:|:---:|:---:|:---:|:---:|:---:|:---:|
> | Meta-Llama-3-8B | 8192 | 21.9 | 11.4 | 11.9 | 1.84 | 0.96 |
> |  | 2048 | 16.7 | 6.8 | 7.2 | 2.31 | 0.94 |
> |  | 512 | 15.4 | 5.6 | 6.1 | 2.54 | 0.93 |
> | Llama-2-7B-hf | 8192 | 18.1 | 6.2 | 6.8 | 2.66 | 0.91 |
> |  | 2048 | 13.9 | 4.2 | 4.7 | 2.95 | 0.89 |
> |  | 512 | 12.9 | 3.7 | 4.2 | 3.08 | 0.89 |
> | Qwen2.5-14B | 8192 | OOM | 19.5 | 21.3 | -- | 0.92 |
> |  | 2048 | OOM | 14.0 | 14.9 | -- | 0.94 |
> |  | 512 | OOM | 12.6 | 13.5 | -- | 0.93 |
>
> 3. **Inference on NVIDIA A100**: Our intention with ResQ was to target inference on consumer devices, which is why we report speedups on an RTX 3090 at batch size 1. While datacenter-scale 4-bit LLM inference is valuable future work, we also provide latency results for Meta-Llama-3-70B on NVIDIA A100 server (Table 7 below). Notably, **ResQ enables the 70B model to fit on a single GPU**, whereas the FP16 baseline requires three GPUs. In this setting, ResQ runs data-parallel inference, while FP16 uses model parallelism.  ResQ achieves up to 4.98× lower latency across various batch sizes and sequence lengths. More sophisticated model parallel inference approaches like pipeline parallelism will only improve throughput of FP16 baseline but will not improve per batch latency.
>
> Table 7: Meta-Llama-3-70B inference latency (in ms) on 3x NVIDIA A100 GPUs.
> | batch\_size | seq\_len | FP16 | ResQ | Improv. |
> |:---:|:---:|:---:|:---:|:---:|
> | 3 | 10240 | 4242 | 20783 | 4.90x |
> | 3 | 8192 | 3361 | 16373 | 4.87x |
> | 3 | 4096 | 1609 | 7871 | 4.89x |
> | 3 | 2048 | 806 | 3888 | 4.82x |
> | 6 | 2048 | 1560 | 7733 | 4.96x |
> | 9 | 2048 | 2309 | 11493 | 4.98x |

---

> > ### Comment · Reviewer_vpAp · 2025-04-08
> >
> > Thanks to the authors for their rebuttal, which has addressed all my concerns. **Overall, I think it is an impressive paper and should definitely be accepted**.

---

### Official Review · Reviewer_UnBL · 2025-03-16

**Overall Recommendation:** 3

**Summary:**

This paper proposes ResQ, a post-training quantization (PTQ) framework that targets aggressive 4-bit quantization of large language models (LLMs) for weights, activations, and KV caches. The key idea is to identify and preserve a low-dimensional subspace of “important” activation components in higher bit precision (8-bit) while quantizing the remaining channels to 4-bit. Specifically, the method uses PCA to find the top-r principal directions of the activation distribution (one-eighth of the hidden dimension in many cases) and keeps those in 8-bit; the other channels go to 4-bit. Within each subspace (high-precision and low-precision), ResQ applies a random orthogonal rotation to suppress outliers. This approach aims to minimize overall quantization error, maintain near-baseline performance at 4-bit, and deliver competitive speedups over 16-bit inference. The authors integrate their projection matrices into the model architecture for minimal runtime overhead and benchmark ResQ extensively on multiple LLMs (Llama series, Qwen2.5, and Qwen2-VL) across language modeling, reasoning, and multimodal tasks. Results show perplexity and speedup gains with minimal additional calibration effort compared to other methods.

**Claims And Evidence:**

ResQ’s key proposition is that it can enable robust 4-bit quantization of weights, activations, and KV caches, largely closing the performance gap to 16-bit baselines. The authors support this by benchmarking on tasks such as Wikitext perplexity and MMLU zero-shot accuracy, showing clear improvements compared to established PTQ methods (e.g., GPTQ, SpinQuant). While the results strongly indicate that ResQ maintains accuracy in low-bit regimes, a gap remains in exploring full multi-GPU or large-batch deployment. Another core claim is that retaining a small low-rank subspace in higher precision is theoretically near-optimal. This is justified by a theoretical bound positing that the directions with largest eigenvalues dominate quantization error, and rotating activations in each subspace helps suppress outliers. However, it relies on assumptions of near-Gaussian distributions, which may not always hold under real-world activation behaviors. A third claim addresses ResQ’s runtime overhead, contending that by fusing projection matrices into existing layer weights, the impact on throughput is minimal. Indeed, single-block timing tests on an RTX 3090 GPU show up to 3× speedup over 16-bit baselines with only minor slowdowns relative to purely INT4 kernels. Yet, the paper does not provide extensive breakdowns of concurrency or other (or multi-) GPU scenarios, leaving some open questions about overhead at scale. Lastly, ResQ’s generalization to large and multimodal models is highlighted by successful application to 70B parameter Llama and Qwen2 VL, although overhead details for extremely large or specialized models are not examined in depth.

**Essential References Not Discussed:**

None

**Experimental Designs Or Analyses:**

The authors primarily employ Wikitext perplexity and multiple 0-shot benchmarks—MMLU, ARC, BoolQ, HellaSwag, and more—to evaluate how well quantized models retain their reasoning or language modeling capabilities. They also include a set of generation tasks such as GSM8K math problems and code completion, offering a more comprehensive view of performance beyond classification or short-answer tasks. ResQ is compared against well-known baselines (e.g., GPTQ, SmoothQuant+, SpinQuant), and the consistent outperformance on perplexity and accuracy indicates that subspace-based 4-bit quantization indeed preserves essential model quality. The authors further examine a small but crucial set of ablations, for instance by removing or altering projection matrices for attention and feedforward blocks, and see noticeable drops in performance when these projections are omitted. Finally, while they do measure kernel-level speedups on a single GPU, there is comparatively limited exploration of multi-GPU scaling or diverse inference conditions. This narrower hardware focus, although still useful, suggests that additional profiling under larger-batch or distributed-serving scenarios would strengthen the paper’s overall claims regarding real-world usability.

**Methods And Evaluation Criteria:**

ResQ fits within the broader family of mixed-precision post-training quantization methods. The authors evaluate primarily on:

- Wikitext Perplexity: A common measure of pure language modeling fidelity.
- 0-shot accuracy on standard reasoning tasks (ARC, BoolQ, etc.) and MMLU to test knowledge retention and general understanding.
- Generative tasks (GSM8K for math, code completion, summarization) to assess the approach on auto-regressive generation.
- MMMU for multimodal comprehension using Qwen2-VL.

Such a diverse evaluation set is a strength: it shows that ResQ is robust across typical PTQ tasks (language modeling, reasoning, generation) as well as specialized tasks (multimodal). However, as with many PTQ papers, the chosen tasks mostly focus on correctness or perplexity rather than fine-grained analysis of speed–accuracy trade-offs under real-world concurrency or large-batch scenarios.

**Other Comments Or Suggestions:**

None

**Other Strengths And Weaknesses:**

**Strengths**
- Good empirical results: Achieves near-best perplexities, 0-shot accuracies, and generative outcomes at 4-bit.
- Implementation details: The authors describe how to fuse the projection matrices into the model to reduce overhead. They also measure actual runtime speedups, reinforcing practicality.
- Applicability: The method is tested on a broad set of LLMs (1B to 70B+ parameters) and even vision-language models, suggesting decent generalizability.

**Weaknesses**
- Limited multi-GPU or concurrency analysis: Real-world inference typically runs parallel requests or large batches. The paper focuses on single-GPU, small-batch speed tests.
- Random rotation overhead: While somewhat mitigated by fusing, there remains an unquantified overhead in larger contexts or many decoding steps, especially for the UD projection in the FFN.
- Minimal exploration of extremely large contexts: Since KV cache quantization is a key selling point, more exhaustive or real-time scenarios with 8K+ tokens might better showcase memory/time improvements.
- Heuristic theory: The analysis of normal-distribution-based error bounds is fairly standard in modern quantization, but it might under- or overestimate the actual distribution complexities in LLM activations.

**Questions For Authors:**

- Have you tested how ResQ’s overhead scales in multi-GPU or distributed settings where activation projections might incur extra synchronization costs? Are the speedups consistent with single-GPU results?
- For models that support context lengths beyond 8k tokens, how does ResQ perform in practice? Does the overhead of repeated rotation or subspace decomposition increase, or does it remain relatively stable?
- You allow a flexible subspace dimension (e.g., d/8). Have you tested intermediate ranks or adaptive ranks per layer? How does the rank selection differ among different layers or for different activation distributions?
- Could combining PCA-based subspace extraction with advanced codebook-based or cluster-based quantization (similar to AQLM) yield further gains? Or is the advantage of random rotation overshadowed by codebook overhead?
- Since you observed that small calibration sets (128–512 samples) can suffice, do you see performance fluctuations if the calibration set is domain-specific vs. general Wikitext? Are there domain mismatches that degrade performance?

**Relation To Broader Scientific Literature:**

ResQ extends a line of LLM PTQ research focusing on 4-bit quantization with minimal accuracy loss:

- GPTQ introduced Hessian-based weight-only quantization.
- SmoothQuant, OmniQuant addressed outlier channels through amplitude scaling.
- QUIK, QuaRot, SpinQuant used outlier or rotation-based methods to handle activation extremes.

**Theoretical Claims:**

The authors prove an upper bound on quantization error under Gaussian assumptions, showing that the subspace-based approach is near-optimal in minimizing the Frobenius norm difference. Although typical in quantization research, the proof remains partly heuristic (due to normality assumptions), but this is consistent with contemporary PTQ literature.

---

> ### Author Rebuttal · Authors · 2025-03-31
>
> Thank you, Reviewer UnBL, for your effort in reviewing our paper and acknowledging the strong empirical results and applicability of ResQ. We answer the listed questions below and incorporate your constructive feedback to further strengthen our work.
> 1. **Heuristic theory:** We agree with the reviewer that most quantization error analyses, including ResQ, assume normally distributed activations. In practice, LLM activations deviate from normality. However, in ResQ, the activation distribution becomes approximately Gaussian after projection via the orthogonal matrix $U$ (as shown in Lemma 4.1). To support this, we compute the kurtosis of activations before and after projection, shown in Table 1 below. Post-projection activations $XU_l$ and $XU_h$ exhibit kurtosis near 3, indicating Gaussianity. Thus, our theoretical assumptions hold in practice.
> Table 1: Kurtosis of Activations before and after projection.
> |Model|Layer|$X$|$XU_l$|$XU_h$|
> |---|:---:|:---:|:---:|:---:|
> |Qwen2.5-3B|Attn|91.9$\pm$38.8|3.0$\pm$0.005|2.9$\pm$0.07|
> ||MLP|179.8$\pm$248.3|3.0$\pm$0.004|2.9$\pm$0.07|
> |Qwen2.5-7B|Attn|75.6$\pm$60.5|3.0$\pm$0.002|3.0$\pm$0.02|
> ||MLP|164.7$\pm$243.1|3.0$\pm$0.0|2.9$\pm$0.04|
> |Meta-Llama-3-8B|Attn|37.9$\pm$50.3|3.0$\pm$0.0|3.0$\pm$0.02|
> ||MLP|6.6$\pm$1.4|3.0$\pm$0.0|3.0$\pm$0.02|
> 2. **Distributed Inference setting :**  Please refer to Table 7 in comment to Reviewer vpAp.
> 3. **Long Context Inference:** We provide per decoder speedup (similar to Fig. 5 in paper) for longer context lengths upto 20000 on Qwen2.5-32B and Meta-Llama-3-70B models in Table 2 below. With increasing sequence lengths, the improvements of ResQ slightly reduces still achieving 2.33x (2.02x) for Qwen2.5-32B (Meta-Llama-3-70B) speedup at sequence length of 20k.
>
> Table 2: Per decoder speedup of long context inference on RTX 3090.
> |Model||||Seq\_len ||||
> |:---:|:---:|:---:|:---:|:---:|:---:|:---:|:---:|
> |Model|8k|10k|12k|14k|16k|18k|20k|
> |Qwen2.5-32B|2.76|2.68|2.58|2.50|2.43|2.38|2.33|
> |Meta-Llama-3-70B|2.36|2.21|2.17|2.12|2.07|2.04|2.02|
>
> 4. **Flexible Subspace Dimension:** It is possible to assign different ranks across layers for $U_B$ and $U_C$ projections since they are different for each layer. It is not possible to have different ranks for $U_A$ with the current approach since it is shared across layers. We test one approach of assigning different rank of 8-bit component to $U_B$ and $U_C$ based on the eigen value distribution of key and value. Specifically, for layers with higher eigen values, we keep 15.6\% channels in 8-bit while for the rest we keep 9.3\% channels in 8-bit achieving 12.5\% high precision components in average. Results with such an approach (shown in Table 3 below) show that flexible rank improves the performance on reasoning accuracy but performs worse on MMLU demanding important future exploration.
> Table 3: Comparison of ResQ with a variant of ResQ having
> flexible rank of $U_B, U_C$.
> |Model|Method|Wiki.PPL|Avg.Reasoning acc.|Avg.MMLU acc.|
> |:---:|:---:|:---:|:---:|:---:|
> |Llama-3.2-3B|ResQ|8.8|59.0|**49.8**|
> ||ResQ-flex.rank|**8.7**|**59.2**|49.0|
> |Meta-Llama-3-8B|ResQ|7.1|63.9|**57.2**|
> ||ResQ-flex.rank|**7.0**|**64.3**|56.8|
> |Qwen2.5-3B|ResQ|**9.0**|61.1|61.2|
> ||ResQ-flex.rank|**9.0**|**61.9**|**61.3**|
> |Qwen2.5-7B|ResQ|8.2|65.3|**69.0**|
> ||ResQ-flex.rank|**8.0**|**65.4**|68.6|
>
> Task-wise results can be found [here](https://shorturl.at/KTiZR) in Table 1.
>
> 5. **Codebook-based quant:** Codebook-based non-linear quantization methods (e.g., AQLM mentioned by the reviewer) focus solely on weight quantization, whereas ResQ takes a more holistic approach by quantizing weights, activations, and KV cache. To the best of our knowledge, no existing work explores non-linear quantization of LLM activations. This is an interesting direction, and we leave it for future work. Thank you for bringing AQLM to our attention; we will cite the work.
>
> 6. **Calibration Dataset:** To analyse the impact of calibration datasets we evaluate performance of ResQ on 3 different datasets. These include two out of distribution language modeling datasets C4 and PTB and one instruction tuning dataset Alpaca. The results provided in Table 4 below shows no significant performance fluctuations with different calibration datasets.
>
> Table 4: Performance of ResQ with different calibration datasets.
> |Model|Calib.dataset|Wiki.PPL|Avg.Reasoning acc.|Avg.MMLU acc.|
> |:---:|:---:|:---:|:---:|:---:|
> |Llama-3.2-3B|Wikitext|**8.8**|59.0|**49.8**|
> ||C4|**8.8**|**61.7**|48.6|
> ||PTB|**8.8**|59.1|47.6|
> ||Alpaca|**8.8**|58.9|48.0|
> |Meta-Llama-3-8B|Wikitext|**7.1**|63.9|57.2|
> ||C4|**7.1**|**64.0**|57.2|
> ||PTB|**7.1**|63.9|56.3|
> ||Alpaca|**7.1**|63.9|**57.5**|
> |Qwen2.5-3B|Wikitext|**9.0**|61.1|**61.2**|
> ||C4|**9.0**|60.7|59.6|
> ||PTB|9.1|59.7|59.5|
> ||Alpaca|**9.0**|61.3|60.9|
> |Qwen2.5-7B|Wikitext|8.2|65.3|**69.0**|
> ||C4|8.2|65.7|68.5|
> ||PTB|**8.0**|65.3|68.8|
> ||Alpaca|8.9|65.7|68.4|
>
> Task-wise results can be found [here](https://shorturl.at/KTiZR) in Table 2.

---

### Decision · Program_Chairs · 2025-05-01

**Decision:**

Accept (spotlight poster)

**Comment:**

The paper introduces ResQ, a post-training quantization (PTQ) which targets weight, activation, and KV-cache tensors for quantization.  Novelly, ResQ applies mixed-precision, targeting a low-rank subspace of critical (high-variance) activation components which are kept in 8-bit precision, while remaining components are aggressively quantized to 4-bit precision.  Invariant random rotation is subsequently applied to further suppress outliers.   The paper theoretically proves that this mixed-precision scheme is theoretical near-optimal, assuming normality (which is standard in LLM quantization, e.g., NF4).  The paper extensively considers the Llama-3 (Llama-3-8B, Llama-3-70B, Llama-3.2-1B, Llama-3.2-3B) and Qwen 2.5 (Qwen2.5-3B, Qwen2.5-72B) models, benchmarks (Wikitext perplexity, average 0-shot common sense reasoning, 0-shot MMLU accuracy, 5-shot GSM8K performance, and LongBench performance), and SOTA PTQ methods ( (e.g., GPTQ, SmoothQuant+, SpinQuant), showing that ResQ significantly outperforms other methods at retaining overall performance.  Finally, the paper demonstrates 1.61 to 3.0 speedups compared to 16-bit inference, with minor slowdowns compared to native int4 inference.

The majority of the reviewers agree that this is a high-quality paper, that both empirical and theoretical results are impressive, and that the novelty of ResQ is an interesting innovation over recent SOTA PTQ methods (e.g., SpinQuant).  Reviewers had several concerns, e.g., Reviewer UnBL questioned the appropriateness of normality for the main theoretical results, Reviewer vpAp questioned the depth of the fused kernel contributions and inclusion of higher-end GPUs (i.e., A100s), and Reviewer UnBL requested a direct 4-bit (non-mixed precision) comparison to other SOTA PTQ methods.  The authors did well to address all criticisms, resulting in raised reviewer ratings (I look forward to the inclusion of these additional experiments in the camera ready).  Reviewer Azbj initially voted in favor of rejection, however, the authors addressed these concerns in the rebuttal and the reviewer did not subsequently engage in further discussion with either the AC-reviewers or authors (for further clarifications).